# Role of the stratospheric chemistry-climate interactions in the hot climate conditions of the Eocene

Sophie Szopa[1], Rémi Thiéblemont[1], Slimane Bekki [2], Svetlana Botsyun[1*], Pierre Sepulchre[1]

[1]Laboratoire des Sciences du Climat et de l'Environnement, LSCE/IPSL, CEA-CNRS-UVSQ, Université Paris-Saclay, Gif-
sur-Yvette, France.
[2]Laboratoire Atmosphère, Milieux, Observations Spatiales, Institut Pierre Simon Laplace, LATMOS/IPSL, CNRS-UVSQ-
Sorbonne Université, Guyancourt  and Paris, France.
*Now at Department of Geosciences, University of Tübingen, Germany

Draft preparation for *Climate of the Past*

*Correspondence to*: Sophie Szopa (sophie.szopa@lsce.ipsl.fr)

## Abstract

The stratospheric ozone layer plays a key role in atmospheric thermal structure and circulation. Although stratospheric ozone distribution is sensitive to changes in trace gases concentrations and climate, the modifications of stratospheric ozone are not
usually considered in climate studies at geological time scales. Here, we evaluate the potential role of stratospheric ozone chemistry in the case of the Eocene hot conditions. Using a chemical-climate model.,we show that the structure of the ozone layer is significantly different under these conditions ($4xCO_2$ climate and high concentrations of tropospheric $N_2O$ and $CH_4$). The total column ozone (TCO) remains more or less unchanged in the tropics whereas it is found to be enhanced at mid and high latitudes. These ozone changes are related to the stratospheric cooling and an acceleration of stratospheric Brewer-
Dobson circulation simulated under Eocene climate. As a consequence, the meridional distribution of the TCO appears to be modified, showing particularly pronounced mid-latitudes maxima and steeper negative poleward gradient from these maxima. These anomalies are consistent with changes in the seasonal evolution of the polar vortex during winter, especially in the Northern Hemisphere, found to be mainly driven by seasonal changes in planetary wave activity and stratospheric wave-drag. Compared to a preindustrial atmospheric composition, the changes in local ozone concentration reach up to 40%
for zonal annual mean and affect temperature by a few Kelvins in the middle stratosphere.

As inter-model differences in simulating the deep past temperatures are quite high, the consideration of atmospheric chemistry, which is computationally demanding in Earth system models, may seem superfluous. However, our results suggest that using stratospheric ozone calculated by the model (and hence more physically consistent with Eocene conditions) instead of the commonly specified preindustrial ozone distribution could change the simulated global surface air
temperature as much as 14%. This error is of the same order as the effect of non-$CO_2$ boundary conditions (topography,

bathymetry, solar constant & vegetation). Moreover, the results highlight the sensitivity of stratospheric ozone to hot climate conditions. Since the climate sensitivity to stratospheric ozone feedback largely differs between models, it must be better constrained not only for deep past conditions but also for future climates.

## 1 Introduction

The absorption of incoming solar ultraviolet (UV) radiation by the stratospheric ozone is responsible for the heating of the stratosphere and hence its dynamical stability. In addition, this absorption is essential to the development of life because it prevents these very harmful UV radiations from reaching the Earth's surface. Stratospheric ozone is thus a key component of the radiative equilibrium and habitability of the Earth (Brasseur & Solomon, 2005). However, although deep time climates are more and more investigated with numerical climate models, the role of the stratosphere in such climates is usually
neglected (e.g. Kageyama et al. 2017, Lunt et al. 2017).

The present-day stratosphere has been intensively studied to understand, anticipate and mitigate the global ozone depletion caused by the emissions of anthropogenic halogenated compounds such as chlorofluorocarbons and halons in the second part of the 20th century (WMO, 2014). The phasing out of the emissions has led to the start of a stratospheric ozone recovery since the end of the 90's (Chipperfield et al., 2017). However, in the context of increasing levels of greenhouse gases (e.g.
GHGs such as $CO_2$, $N_2O$, and $CH_4$) and associated climate changes, the sensitivity of stratospheric ozone to other drivers, especially climate-related drivers, is increasingly investigated. For example, stratospheric ozone is sensitive to changes in $N_2O$, $CH_4$ and water vapour levels. $N_2O$ enters in the stratosphere at the tropical tropopause and controls the levels of NOx, which are the most efficient ozone-destroying radicals in the middle stratosphere. Enhanced $CH_4$ levels increase ozone production in the troposphere and lower stratosphere but also lead to higher water vapor levels, which tend to favour ozone
destruction (Revell et al. 2012, Bekki et al. 2011). An increase in $CO_2$ concentration results in a cooling of the stratosphere, which slows down ozone destruction in the upper stratosphere and hence favours ozone recovery in this region. In addition to stratospheric chemical changes, the ongoing climate change tends to intensify the large-scale stratospheric overturning circulation (the so-called Brewer-Dobson circulation), which is responsible for upward transport of air in the tropics and poleward and downward transport at middle and high latitudes (see e.g. Butchart (2014) and references therein). These
circulation changes result in reduced ozone levels in the tropical lower stratosphere due to the faster ascent of air in the lower tropical stratosphere (Avallone and Prather 1996) and enhanced ozone levels at middle and high latitudes (Bekki et al. 2011). This illustrates how stratospheric ozone responds to climate change. More recently, Chiodo et al. (2018) presented an analysis of the stratospheric ozone layer response to an abrupt quadrupling of $CO_2$ concentration in 4 chemistry climate models. As found previously (see e.g. WMO, 2014), they showed that increased $CO_2$ levels in the 4 models lead to a
decrease in ozone concentration in the tropical lower stratosphere and an increase at high latitudes and in the upper stratosphere. However, there were large differences between models in the magnitude of the ozone response.

At the same time, stratospheric ozone changes also influence the climate. For instance, climate models have to account for the formation of the stratospheric ozone hole to be able to reproduce correctly the trends in Antarctic surface temperatures observed during the last half century (e.g. Son et al., 2010; McLandress et al., 2011). Considering larger climate perturbations, Nowack et al. (2015) performed an abrupt $4xCO_2$ experiment with a comprehensive ocean-atmosphere-chemistry-climate model and found that neglecting stratospheric ozone changes triggered by $CO_2$ increase (i.e. specifying a fixed ozone climatology in the model) led to overestimate the surface global mean temperature response by about 1K (i.e. 20% of the total surface temperature response). Chiodo & Polvani (2017) carried out the same numerical experiment ($4xCO_2$) with a different chemistry-climate model and found that, in contrast to Nowak et al. results, stratospheric ozone changes played a negligible role in the global surface temperature response. Nonetheless, they found that the stratospheric ozone feedback in their model significantly reduced the $CO_2$-induced poleward shift of the mid-latitude tropospheric jet by lowering the strength of the meridional temperature gradient near the tropopause. These results suggest that stratospheric ozone perturbations should be accounted for in climate models in order to fully capture the climate response to GHGs changes.

In the past of the Earth, the oxygenated atmosphere has encountered hot climatic conditions due to strong greenhouse effects. During the early Eocene (~ 56-50Ma) terrestrial temperatures at high latitudes were possibly up to 20 K higher than modern ones (Masson-Delmotte et al. 2013). Under such a warm climate, biogenic emissions of $N_2O$ and $CH_4$ were likely to be drastically boosted, being 4 to 5 times higher than the preindustrial ones (Beerling et al., 2011). Note however that, in most modelling studies of deep time climates the role of non-$CO_2$ gases is neglected (e.g. Deep MIP, Lunt et al. 2017). Beerling et al. (2011) studied the tropospheric chemical composition under a warm climate and potentially high biogenic emissions of the early Eocene (55 Ma). Using an Earth System Model including tropospheric chemistry, they found that the OH concentration, which is the main oxidant for most compounds in the troposphere, was significantly reduced (by 14 to 50%) due to higher levels of compounds to oxidize. The high tropospheric levels of reactive greenhouse gases ($N_2O$, $CH_4$ and $O_3$) were maintained under these conditions. Considering the full Earth system interactions, and in particular albedo change due to melting of continental snow, Beerling et al. (2011) calculated an increase of 1.4 to 2.7K in surface temperatures due to tropospheric chemical composition changes for the Eocene. However, since their model did not include stratospheric chemistry, they could not study stratospheric composition changes. Unger & Yue (2013) investigated the chemistry-climate feedbacks in the mid-Pliocene (~3 Ma) using a vegetation-chemistry-climate model simulating both stratospheric and tropospheric chemistry. This epoch is cooler than the Eocene but still of interest because its global climate is thought to be as warm as the climate projected for the end of the ongoing century (+2-3 K compared to present-day). Compared to preindustrial conditions (PI), the Unger & Yue (2013) model simulations indicated that the mid-Pliocene ozone burden was higher, by 25% in the troposphere and by 5% in the stratosphere. The global stratospheric ozone increase, resulting from a stronger tropical upwelling and a lower ozone destruction in the stratosphere, led to a 20% decrease in tropospheric ozone photolysis and hence OH production. As a consequence, tropospheric OH concentrations were reduced by 20-25% and hence the lifetime and burden of important reactive species (CO, $CH_4$) were significantly increased. Unger & Yue (2013) showed

that the warming effect of the changes in chemically reactive compounds (i.e. $CH_4$, $N_2O$, tropospheric $O_3$) could have represented ~75% of the warming from $CO_2$ increase. The studies of Beerling et al. (2011) and Unger & Yue (2013) suggest that non-$CO_2$ greenhouse gases may have played a significant role in the overall climate in the Cenozoic greenhouse worlds.

As pointed out previously, most studies of Cenozoic paleoclimates assume that the atmospheric composition is fixed except for $CO_2$ because there is no estimate of these composition changes. The purpose of this paper is (i) to investigate, using a stratospheric chemistry-climate model, to what extent the stratosphere, notably the ozone layer, might have been different in the early Eocene conditions and (ii) to estimate the possible effects of these stratospheric changes on the tropospheric oxidizing capacity and climate. The Eocene is characterized by high surface temperatures, elevated $CO_2$ levels, the absence of ice cap and a large extent of tropical vegetation. High $CH_4$ and $N_2O$ levels are also expected based on ESM simulations (Beerling et al. 2011). Whereas the data are sparse and have large uncertainties for the geological past, several proxy-based reconstructions of $CO_2$ levels and surface temperatures have been released, notably allowing to build an harmonised protocol for climate modelling of the early Eocene (Lunt et al. 2017), which will be used to intercompare climate model sensitivity during the on-going Paleoclimate Modelling Intercomparison Project (PMIP4). This protocol gathers recommendations on paleogeography, land cover, $CO_2$ and $CH_4$ concentrations, natural aerosols, solar constant and astronomical parameters but no recommendations have been provided yet for stratospheric conditions. In this work, we propose to examine the role of the stratospheric ozone layer in the Eocene climate. We first investigate the stratospheric ozone response to GHG-induced warm climate such as the one expected under the Eocene conditions and compare it to preindustrial climate conditions. We then discuss the methodology to account properly for these stratospheric ozone changes in deep time paleoclimate simulations. Finally, based on the model simulations, we estimate the difference in UV radiation reaching at Earth surface between this epoch and the preindustrial period and the resulting impact on tropospheric chemistry. The potential climate forcing of the stratospheric changes is also discussed.

## 2. Methodology

### 2.1 The LMDz-REPROBUS climate-chemistry model

Simulations are performed with the stratospheric chemistry-climate model developed in the framework of the IPSL-Earth System Model (IPSL-CM) development (Dufresne et al. 2013). The stratospheric chemistry is computed with the REPROBUS chemical model (Lefèvre et al. 1994, 1998, Jourdain et al. 2008) coupled with the LMDz atmospheric general circulation model (Hourdin et al. 2013). REPROBUS describes the chemistry of stratospheric source gases such as $N_2O$, $CH_4$, $CH_3Cl$, $CH_3Br$, and the associated radical chemistry of hydrogen, nitrogen oxides, chlorine, and bromine species. It computes the global distribution of trace gases, aerosols, and clouds within the stratosphere considering gas-phase and heterogeneous chemistry. The heterogeneous chemistry component takes into account the reactions on sulfuric acid aerosols, and liquid (ternary solution) and solid (Nitric Acid Trihydrate particles, ice) Polar Stratospheric Clouds (PSCs). The gravitational sedimentation of PSCs is simulated as well. The LMDz-REPROBUS chemistry-climate model allows an

interactive coupling of ozone, shortwave heating rates and dynamics as recommended in Sassi et al. (2005). The resolution of the model is 3.75 degrees in longitude x 1.9 degrees in latitude and 39 vertical levels, with around 15 levels above 20 km and around 24 above 10km and a lid height at ~70 km.

LMDz and LMDz-Reprobus have been involved in a large range of studies, model inter-comparisons and evaluations, notably through the participation of the LMDz model to international Coupled Model Intercomparison Project (CMIP, phases 3, 5 and currently 6) and the participation of LMDz-Reprobus to Chemistry Climate Model Validation (SPARC-CCMVal, 2010) and Chemistry Climate Model Initiative (CCMI, Morgenstern et al., 2017). Results presented in the recent studies of e.g. de la Cámara et al. (2016a,b), Thiéblemont et al. (2017) and Ayarzagüena et al. (2018) have shown that the stratospheric chemistry, dynamics and transport simulated by the LMDz model and its CCM version are consistent with satellite observations, reanalysis and other models of the same kind.

## 2.2 Simulation set-up

The setup of the four simulations performed in this study is summarized in Table 1. All the simulations consist of 30-year time slices, starting from atmospheric physical conditions and surface temperatures taken from very long coupled atmosphere-ocean simulations. For the analysis of our chemistry-climate simulations, a 5-year spin-up is considered. For all the simulations, the solar constant is set to $1366 Wm^{-2}$ and orbital parameters (obliquity, precession, and eccentricity) are set to modern values as recommended in the DeepMIP protocol (Lunt et al. 2017). Oxygen variations are poorly constrained over pre-quaternary timescales and there is no consensus on the oxygen variations through the Cenozoic (see Fig. 1 of Wade et al. 2018). In view of these uncertainties, we use a present-day oxygen content to investigate Cenozoic past climates, as commonly done in climate models.

### 2.2.1 Preindustrial simulations

The boundary conditions of our preindustrial experiment (PREIND) include modern topography, land-sea mask, ice sheets and climatological mean values computed over the 1870-1899 period for sea surface temperatures (SSTs) and sea-ice extent. Greenhouse gases are set to preindustrial values, i.e. $CO_2$ level at 285 ppm, $CH_4$ level at 791 ppb, and $N_2O$ level at 275 ppb. Halogenated ozone depleting substances of anthropogenic origin (i.e. fluorocarbons) are set to zero. Naturally emitted halogenated compounds ($CH_3Br$ and $CH_3Cl$) are prescribed to their preindustrial levels (respectively, 7ppb and 482 ppb).

### 2.2.2 Eocene base case simulation

As for the PREIND experiment, the Eocene experiment (EOCENE) includes interactive chemistry, which allows to calculate stratospheric composition. The physical boundary conditions for the EOCENE experiment (e.g. SSTs, sea-ice, land surface properties) are based on a climate simulation done with the fully coupled low-resolution Fast Ocean Atmosphere Model (FOAM; Jacob, 1997) and the Lund-Potsdam-Jena (LPJ) dynamic global vegetation model (Sitch, 2003) coupled offline as illustrated Figure 1. The LPJ-FOAM coupled ocean-atmosphere-vegetation simulation provides the surface conditions

(SSTs, land surface conditions) required to simulate the climate with the LMDz atmospheric general circulation model. FOAM is forced with the Eocene paleogeography reconstruction of Herold et al. (2014). Compared to the present-day paleogeography, it includes major modifications, namely closed Drake and Tasman Seaways, an open Central American Seaway, and an open Parathetys Sea. Topography is altered as well, with lower Tibetan plateau and Andes. $CO_2$ is set to

1120 ppm, equivalent to 4x $CO_2$ preindustrial level ([$CO_2$]$_{PI}$), as Eocene $CO_2$ estimates range between 400 and 2400ppm (as reported by Lunt et al. (2017) based on boron isotopes analysis from Anagostou et al. 2016). This $CO_2$ value lies in the low-end of the Eocene compatible GHG forcing ranges, in particular those recommended by the DeepMIP project, which proposes to test 3x[$CO_2$]$_{PI}$, 6x[$CO_2$]$_{PI}$ and 12x[$CO_2$]$_{PI}$ (Lunt et al. 2017). After 2,000 model years, SSTs simulated by FOAM are averaged over the last 100-year period to build a 12-month (seasonally varying) climatology used as a boundary

condition for LMDz. FOAM coupling with the LPJ vegetation model provides an equilibrated vegetation as well, whose albedo and rugosity are extracted to serve as continental boundary conditions for LMDz. The global mean SST that we use are of 17.3°C for preindustrial and of 23.9°C for Eocene. These values lie in the ranges presented for 4 model realisations in Lunt et al. (2012). These ranges are between 15.2 and 17.9°C for PI and between 22.2 and 26.4°C for Eocene when considering 4x[$CO_2$]$_{PI}$ (Note that when $CO_2$ varies from 2x[$CO_2$]$_{PI}$ to 16x[$CO_2$]$_{PI}$, the range of SST is between 21.4 and

29.7°C) (Lunt et al. 2012). In addition, the meridional surface temperature gradient is of 24.6°C over ocean and 33.7 over land in our protocol when the ranges with the 4x[$CO_2$]$_{PI}$ experiments shown in Lunt et al. (2012) are [24 ; 33]°C  and [25.5 ; 37]°C respectively. Numerous recently published paleoclimate studies are based on the two-step methodology based on FOAM-LPJ and LMDz and this set-up has been shown to perform well (e.g. Botsyun et al. 2019, Ladant et al., 2014; Ladant et al. 2016; Licht et al., 2014; Pohl et al. 2016; Porada et al. 2016).

Applying a coupled vegetation-atmosphere to the particularly warm climate of the early Eocene (55Ma), Beerling et al. (2011) have estimated that $CH_4$ and $N_2O$ concentrations should have been much higher than nowadays and could have lied in the 2384-3614 ppb and 323-426 ppb ranges respectively. The direct climate impact of highly enhanced $CH_4$ and $N_2O$ levels is accounted for by setting $CO_2$ to high level (1120ppm) in the radiative module in our atmospheric circulation model (Table 1). Ozone chemistry is affected by changes in $N_2O$ and $CH_4$ (e.g. Revell et al. 2012). To account for this effect, there are $CH_4$

and $N_2O$ chemically active tracers in the REPROBUS chemical model (i.e. modified by the transport and chemistry schemes). Their surface concentrations are taken from the modelling study of Beerling et al. (2011) and $CH_4$ and $N_2O$ surface concentrations are set to 3614ppb and 323ppb respectively in the chemistry module (REPROBUS). Their global distributions change with time during a simulation, but they are not used as inputs to the radiative scheme and hence their changes do not affect the climate, only ozone changes do.

**2.2.3 Eocene simulations with prescribed stratospheric ozone**

In addition to the EOCENE experiment in which stratospheric ozone is calculated interactively, two other Eocene simulations (EOCENE_OzRoyer, EOCENE_Oz1855) are performed in which different climatological ozone representations are specified instead of ozone being calculated interactively. The ozone climatology in the EOCENE_OzRoyer experiment is

rather typical of the 1980's ozone distribution. It originates from fits to the ozone profile from Krueger and Mintzner (1976) and variations with altitude and latitude of the maximum ozone concentrations and total column ozone from Keating and Young (1986). This OzRoyer ozone climatology was constructed by J.-F. Royer (CNRM, Meteo France) and implemented in the LMDz atmospheric circulation model in the 1980's. The Oz1855 ozone climatology in the EOCENE_Oz1855 experiment is more representative of the preindustrial period. It is based on a 11-year mean climatology centered on 1855 derived from historical transient LMDz-REPROBUS simulations (Szopa et al. 2013). This ozone climatology is commonly used for the simulation of past climates with the LMDz model.

The comparison between EOCENE and PREIND experiments, which both include interactive chemistry of the stratosphere, allows us exploring and quantifying the impacts of Eocene warm climate on stratospheric circulation and composition (Section 3). Furthermore, by comparing EOCENE experiment with EOCENE_OzRoyer and EOCENE_Oz1855 experiments allows to assess the role of the stratospheric ozone representation on the climate response to Eocene extreme conditions (Section 4). Note that the statistical significance of anomaly fields is estimated here using a Student's t-test.

## 3. Impacts of Eocene conditions on stratosphere

### 3. 1 Stratospheric ozone in Eocene conditions

We first investigate the impact of Eocene conditions on stratospheric ozone with respect to preindustrial conditions. Figure 2a,b shows the latitude/pressure zonally average cross-sections of temperature and ozone anomalies associated with the Eocene conditions. As expected, the $CO_2$ increase leads to a global radiative cooling of the stratosphere with decreases in temperatures exceeding 12K above 10 hPa (~32 km) and a warming of the troposphere (Figure 2a). In the troposphere, we further notice a more pronounced Antarctic amplification of the temperature signals, which contrasts with present-day climate conditions where a more pronounced Arctic amplification of the global warming is observed (IPCC, 2013). This signal, consistently simulated by several models (Lunt et al., 2012), is linked to the absence of Antarctic ice sheet in Eocene boundary conditions, that leads to lower surface topography and albedo. The cooling of the stratosphere slows down the ozone destruction, resulting in an increase in stratospheric ozone concentrations (Haigh and Pyle, 1982). This is consistent with the statistically significant positive ozone anomalies found above 50 hPa (~20 km) over the polar regions and above 10 hPa in the tropical band (Figure 2b). Note that this effect increases with altitude in the stratosphere as the photochemical control on ozone level becomes prominent (Brasseur and Solomon, 2005). Similarly to the results of Chiodo et al. (2018), which investigate the effect of quadrupling $CO_2$ by starting from preindustrial climate, a maximum ozone increase of ~40% is found at about 2-3 hPa (~40 km). Note that our simulation, the stratospheric chemistry is also modified by the increase of $N_2O$ and $CH_4$. However, their effect only reaches a maximum of 3% in the equatorial upper stratosphere (~5 hPa) (see supplementary Figure S1). Although this chemical effect on ozone is statistically significant, its impact appears to be small compared to the upper stratosphere 40% increase in ozone due to increasing $CO_2$.

In contrast, the lower tropical stratosphere (30°S-30°N, 100-30 hPa) exhibits a statistically significant ozone decrease of up to 40%. In this region, the ozone concentration is mostly controlled by transport processes (Brasseur and Solomon, 2005), especially the strength of the Brewer-Dobson circulation ascending branch. Figure 2c shows the age of air (AoA) calculated after 20 years of simulations by taking as a reference entry point the equatorial lowermost stratosphere, slightly above the tropopause (i.e. pressure level corresponding to 74 hPa). Globally, the stratospheric AoA is younger in the Eocene experiment than in the preindustrial one, revealing an acceleration of the Brewer-Dobson circulation under Eocene conditions. This, in turn, is consistent with the reduced ozone concentration in the lower tropical stratosphere. Note also that the tropopause height is globally lifted up in the Eocene experiment (not shown). The rise of the tropopause is a robust feature of warmer climate conditions (Sausen and Santer, 2003) and contributes to the negative ozone anomaly found in the vicinity of the tropopause region (Figure 2b) (Dietmüller et al. 2014).

Next, we examine anomalies of the annual total ozone column ozone (TCO). Figure 3 shows the comparison of the latitudinal distribution of the annual TCO for Eocene and Preindustrial conditions. In both simulations (Figure 3a), the TCO shows a minimum in the tropical region [20°S, 20°N] of ~270 Dobson unit (DU) and maxima near 55°N and 55°S, followed by poleward decreases that are more pronounced in the Southern Hemisphere. The differences between Eocene and preindustrial conditions (Figure 3b) reveal no changes in the tropical band 20°S-20°N, but statistically significant positive anomalies at mid-latitudes and in polar regions. The mid-latitude maxima reach ~390 DU for preindustrial conditions, whereas they exceed 430 DU for Eocene conditions. This latitudinal distribution of TCO anomalies is overall consistent with projections of TCO anomalies simulated in response to the 21[st] climate change post-CFC era (Li et al., 2009) or to an abrupt $4xCO_2$ increase from preindustrial climate conditions (Chiodo et al., 2018). The detailed comparison of our results with those of Chiodo et al. (2018) shows, however, noticeable differences at high latitudes. In our simulations, the TCO anomalies peak at 45° S and 50° N with maximum differences of 50 and 60 DU, respectively; the anomalies decrease from these maxima to about 30 DU at high latitudes (Figure 3b). In Chiodo et al. (2018), hints of such a decrease was found in the Southern Hemisphere for only two out of the four models that are inter-compared, and no such decrease was seen in the Northern Hemisphere. The negative poleward TCO gradient at high latitudes appears to strengthen markedly under Eocene conditions in the Northern Hemisphere (Figure 3l). The seasonal dependence of the TCO high-latitudes poleward gradient for the Eocene and preindustrial conditions in the Northern Hemisphere is explored on Figure 4. Figure 4 reveals that, under Eocene conditions, the negative gradient is particularly pronounced during the winter season (from October to March), when the stratospheric polar vortex dominates the high latitudes circulation in the Northern Hemisphere. Hence, this indicates substantial changes in the stratospheric circulation associated with the Eocene conditions that we examine further in section 3.2.

**3. 2 Seasonal evolution of the Northern Hemisphere stratospheric polar vortex in Eocene conditions**

An overview of the annual average background zonal circulation in preindustrial conditions and its anomalies associated with Eocene conditions is shown on Figure 5. In the Eocene conditions, the high latitudes stratospheric westerlies maxima, indicative of the average location of the core of the stratospheric Southern and Northern Hemisphere polar night jets (near 60° S and 60° N), appear to be overall stronger and also shifted equatorward in comparison with the preindustrial climatology (black contour). These results hence suggest a strengthening and an extension of stratospheric polar vortices, which develop during winter in each Hemisphere. Note also that the upward extension of the subtropical upper tropospheric jets in both hemispheres (centred near 35°N/S around 200 hPa) are consistent with the tropopause rising associated with Eocene conditions.

To identify the processes leading to the strengthening of the stratospheric polar vortex under Eocene conditions, we explore the stratospheric dynamical wintertime evolution in the Northern Hemisphere, where the largest changes are found in our simulations (e.g. Figure 5). Figure 6 shows the monthly evolution of the zonal-mean zonal wind from October to March in the Northern hemisphere. Regardless of simulated climate conditions, the winter season in the stratosphere is characterized by the development of a mid-to-high latitudes strong westerly jet (or polar night jet - the center of which roughly corresponding to the edge of the polar vortex), which maximizes in mid-winter (December-January). In early and mid-winter (Figure 6a-e), the polar night jet in Eocene conditions appears however to be twice as strong as in preindustrial conditions as shown e.g. in January (Figure 6d) where the maximum anomaly near 60°N/5hPa (~40 m/s) associated with Eocene conditions is larger than the preindustrial climatology (~30 m/s). Such a strong boreal polar night jet was also found in Eocene simulations of Baatsen et al. (2018). In late winter (Figure 6f), the differences in polar night jet strength between the two experiments are no longer statistically significant in the middle stratosphere and appear to be even reversed in the upper stratosphere; i.e. an easterly anomaly is found near the stratopause at mid-latitudes. This indicates a very fast decay of the polar vortex in Eocene conditions in late winter (see also supplementary Figure S1). These differences in the seasonal evolution of the zonal-mean zonal wind are consistent with the seasonal evolution of the ozone gradient shown on Figure 4. Indeed, under Eocene conditions, the stronger winter polar vortex is associated with a reinforcement of the mixing barrier at its edge. This leads in turn to a reduction of air exchanges between mid- and polar latitudes and hence to a steepening of the poleward ozone gradient. Similar (though less pronounced) differences between Eocene and preindustrial conditions are found in the Southern Hemisphere (not shown).

At first glance, the strengthening of the polar vortex under Eocene conditions may seem in contradiction with the global acceleration of the Brewer-Dobson circulation as diagnosed by the younger stratospheric age of air (Figure 2c). Indeed, a faster Brewer-Dobson circulation is associated with a stronger planetary wave drag in the stratosphere (i.e. an enhanced wave breaking), which in turn should lead to a weaker polar vortex. In the following, we hence investigate the seasonality of the planetary stationary wave activity and its interaction with the mean flow by calculating the Eliassen-Palm flux (hereafter EP-flux) divergence, here scaled to units of zonal acceleration (Andrews et al., 1987):

$$divEP = \frac{\vec{\nabla} \cdot \vec{F}}{\rho_0 a \cos \phi}$$

where $\vec{F}$ is the EP-flux whose components are

$$F^{(\phi)} = \rho_0 a \cos \phi \left( \overline{u_z} \frac{\overline{v'\theta'}}{\overline{\theta_z}} - \overline{v'u'} \right)$$

$$F^{(z)} = \rho_0 a \cos \phi \left( \left[ f - \frac{(\bar{u} \cos \phi)_\phi}{a \cos \phi} \right] \frac{\overline{v'\theta'}}{\overline{\theta_z}} - \overline{w'u'} \right)$$

$f$ is the Coriolis parameter, $a$ is the Earth's radius, $\theta$ is the potential temperature, $\rho_0$ is the density profile of the atmosphere and $(u, v, w)$ are the three-dimensional velocity components in spherical coordinates $(\lambda, \phi, z)$, where $z$ is the log-pressure. Overbars indicate zonal-mean and primes denote departure from zonal mean. As shown by Edmon et al. (1980), the EP-flux constitutes a measure of the Rossby wave propagation from one height ($z$) and latitude ($\phi$) to another and its divergence (divEP) gives information about the forcing of the mean flow by the eddies.

Figure 7 displays the monthly evolution in winter of the EP-flux and its divergence for the preindustrial conditions experiment. This analysis shows that, throughout winter, the wave activity penetrates the stratosphere (as indicated by the vectors) near 55°N, propagates upward and tends to be increasingly refracted toward the equator with height. The dissipation of planetary waves exerts a westward momentum forcing on the mean flow between 30°N and 70°N (as diagnosed by the Eliassen-Palm flux convergence), which maximizes along the equatorward flank of the polar night jet where planetary wave breaking is large. This contributes to erode and weaken the polar vortex, to a warming of the polar stratosphere and drives a persistent poleward mass transport in order to conserve the angular momentum. By mass continuity, this induces an upward transport at low latitudes and an extratropical downwelling (hence driving the Brewer-Dobson circulation). Under preindustrial climate conditions, we note that the wave activity and its interaction with the mean flow peaks in December/January (Figure 7c,d) but is already large in November (Figure 7b). Therefore, this contributes to slow down the radiatively-driven development of the polar vortex in early winter.

As shown on Figure 8, under Eocene conditions, it appears that the planetary wave activity penetrating the stratosphere in early winter (i.e. November-December, Figure 8b,c) is significantly reduced and deflected equatorward as revealed by the downward and equatorward pointing of the EP-Flux vector in the lower mid-latitude stratosphere. This is associated with an anomalous positive E-P flux divergence (i.e. a reduced convergence) throughout the depth of the stratospheric polar night jet (near 60°N), which indicates a substantially reduced westward momentum forcing by planetary waves and hence allows a stronger development of the polar vortex in early winter in comparison with preindustrial conditions. In contrast, from January (Figure 8d), the planetary wave activity becomes significantly larger under Eocene conditions, the westward forcing appears to be strongly amplified in the upper stratosphere and to progressively propagates downward in February (Figure 8e). This is consistent with the reversal of the zonal mean zonal wind anomaly in the upper stratosphere, but also with the overall extremely rapid deceleration of the polar vortex strength noted previously (see Figure 6 and Figure S2). In addition, we analyzed the residual mass circulation (not shown) derived from the transformed Eulerian-mean formalism (Andrews et

al., 1987). While no clear changes in the strength of the Brewer-Dobson circulation are found in early winter between Eocene and preindustrial conditions, late winter (February-March) reveals an important acceleration in Eocene conditions, which is consistent with the much stronger wave forcing found throughout the extratropical stratosphere (Figure 8e,f). These results are consistent with a net acceleration of the Brewer-Dobson under Eocene conditions in comparison with preindustrial conditions as revealed by the younger age of air (Figure 2c). Note that the Brewer-Dobson acceleration appears to be more pronounced in the Northern Hemisphere, where the mean flow and wave activity anomalies are found to be stronger than in the Southern Hemisphere (not shown).

Although the large changes in the background state stratospheric circulation and its seasonal evolution under Eocene conditions in comparison with preindustrial climate conditions appear to be largely wave-driven, the origin of the identified changes in the planetary wave activity and its interaction with the mean-flow remains to be determined. Note that this does not uniquely depends on changes in tropospheric wave sources, but also on changes in the background flow itself, which modulates the wave propagation and the nature of wave-mean flow interactions. The planetary wave activity entering the stratosphere can be altered by numerous factors such as sea-surface temperature changes (e.g. Hu et al., 2014), sea-ice changes (e.g. Kim et al., 2014), wind changes near the tropopause (e.g. Shepherd and McLandress, 2011; Karpechko and Manzini, 2017) or topography (Shi et al., 2014). Additional simulations of Eocene and preindustrial with the atmospheric model (LMDz) without interactive chemistry and with a flat topography reveals that changes in the topography have first order effects on planetary wave activity and hence on the stratospheric dynamics (not shown). Between the Eocene and the preindustrial conditions, beside large changes in the topography, important changes in air-sea thermal contrasts, sea-ice cover and sea surface temperature could all have a substantial influence on stratospheric circulation. The complexness of these effects and their possible interactions make an unambiguous attribution impossible in the absence of a devoted experimental protocol, that is out of the scope for the present study.

## 4. Climate impact of an interactive stratospheric chemistry

Model results shown in the previous section suggest that stratospheric dynamics and composition were very significantly altered under Eocene hot conditions in comparison with preindustrial climate conditions. In turn, these stratospheric changes may also have influenced the establishment of the Eocene climate. Nowack et al. (2015) have shown that stratospheric changes driven by ozone changes can have an impact on the climate sensitivity in the context of high GHG concentrations for present day conditions. The importance of this stratospheric ozone-climate feedback has, however, not been assessed in the context of Eocene hot climate. In this section, we estimate the role of this feedback on the overall Eocene climate response by comparing the EOCENE experiment (i.e. where ozone is calculated interactively and, hence, is physically consistent with Eocene conditions) with Eocene_OzRoyer and Eocene_Oz1855 experiments (i.e. where preindustrial ozone climatologies are prescribed in Eocene simulations). The latter simulations follow the protocol usually recommended for simulating paleo climates (e.g. Kageyama et al., 2017).

Table 2 shows the total ozone and temperature changes as well as the effective radiative forcings induced by the use of an interactive stratospheric chemistry instead of seasonally varying prescribed climatologies. All the results in this section are discussed in term of 25-year mean. The effective radiative forcing is computed as the difference of net radiative flux at the top of atmosphere (TOA) between two atmospheric simulations (with ozone calculated interactively – with ozone climatology) as defined in Fig 8.1.d of Myhre et al. (2013).

Figure 9 shows the distribution of total column ozone as a function of latitude for the different configurations. The preindustrial ozone distribution computed by REPROBUS and the Szopa et al. (2013) ozone climatology are represented as well. Comparing only the different preindustrial ozone distributions, the TCO in the experiment with interactive calculation of ozone (PREIND in black) is higher than those of the climatologies (OzRoyer in blue, Oz1855 in brown). In addition, the interactive calculation of Eocene ozone (EOCENE in red) leads to much higher TCO than in the preindustrial climatologies (blue, brown), the 2000 climatology (green) and the preindustrial interactive ozone simulation (black). The global mean TCO is increased by about 45 DU or 35 DU with respect to the preindustrial OzRoyer or Oz1855 climatologies respectively. For the sake of comparison, the global mean TCO had decreased by about 13 DU only between the 1960's and the end of 90's because of the past emissions of anthropogenic halogenated compounds, and the expected TCO increase at the 2100 horizon is projected to be between 13 and 32 DU depending on future anthropogenic emissions of GHGs (Bekki et al. 2013, Szopa et al. 2013). Taking the ozone distribution calculated in the EOCENE simulation as the reference, the TCO bias in an inappropriate ozone climatology can be 2 times higher (in the case of the OzRoyer climatology) than the TCO change calculated by the model between Eocene and preindustrial conditions (EOCENE versus PREIND, see section 3).

The TCO difference between the EOCENE ozone distribution and ozone climatologies peak at mid latitudes (about 40°), reaching almost 70 DU for the Oz1855 climatology and about 100 DU for the OzRoyer climatology (Fig. 10a, left). In order to identify the regions responsible for the general increase in TCO calculated from preindustrial to Eocene conditions, the zonal mean distribution of ozone difference between EOCENE and Oz1855 climatologies are plotted in DU/km on the Fig. 10b (right panel). The TCO increase in EOCENE is largely due to an enhancement in ozone in the upper stratosphere. The TCO change in the tropics are very moderate because the upper-stratospheric ozone enhancements are more or less compensated by lower stratospheric ozone decreases brought about by the acceleration of the Brewer-Dobson circulation, namely the faster ascent in the tropics (section 3). TCO enhancements peak at mid-latitudes because the ozone concentration increases reach down to the 150 hPa at mid-latitudes, again certainly linked to the acceleration of the Brewer-Dobson circulation and more specifically the faster descent at mid- and high latitudes. Below 200 hPa, around the tropopause region, extra-tropical EOCENE ozone concentrations are lower than in Oz1855 climatology, mostly because of the rise in the tropopause height (section 3).

Ozone changes naturally lead to temperature changes, especially in the stratosphere where ozone and temperature are closely coupled. Figure 11 shows the zonal mean distribution of temperature difference between EOCENE and EOCENE_Oz1855 simulations. The impact is weak below 400 hPa since SSTs are fixed and identical in all the Eocene simulations (EOCENE, EOCENE_OzRoyer, EOCENE_Oz1855). The change in zonal mean temperatures below 400 hPa does not exceed 0.15 K,

but can almost reach 0.5 K for the northern polar latitudes when the interactive ozone simulation (EOCENE) is compared to the EOCENE_OzRoyer (not shown). In contrast, temperatures above about 200 hPa are significantly impacted by the choice of ozone distribution used in the model. Temperatures are more than 2.5 to 3 K higher at middle and high latitudes in both hemispheres when ozone is calculated interactively instead of using the preindustrial 1855 climatology. The largest differences are found near the stratopause region (above 5 hPa), in the lower polar lower stratosphere(~130 hPa) and in the middle tropical stratosphere (~60 hPa), where temperatures are respectively higher than 6 K, higher than 4 K and lower than 3.5 K in the simulation with interactive ozone compared to the one with the preindustrial climatology.

## 5. Do we need to consider stratospheric ozone feedback in deep past simulations?

### 5.1 Impact on climate

The consideration of a stratospheric ozone compatible with the Eocene conditions perturbs the radiative balance compared to the use of a preindustrial climatology. The net radiative change (shortwave + longwave) between the simulation with interactive chemistry and the simulation with preindustrial climatology corresponds to a 1.7 $W.m^{-2}$ effective radiative forcing (RF). This radiative forcing results from combined positive RFs in the longwave (LW) and shortwave (SW) simulated in the tropics. Beyond 50°, the positive SW RF is partly counterbalanced by a negative longwave RF (see Table 3 and Figure 12). This radiative forcing from the stratospheric ozone response represents a positive climate feedback, which is commonly ignored in Eocene climate simulations. In order to estimate the potential impact of an interactive ozone on surface temperature under the Eocene conditions, we consider a large range of climate sensitivity from 0.4 to 1.2 $K.W^{-1}.m^{-2}$ (Knutti et al., 2017). Given such a broad range, the surface temperature response to this stratospheric forcing could range from 0.7 to 2.0 K (assuming an effective radiative forcing of 1.7 $W.m^{-2}$). The surface temperature response to a specific radiative forcing depends on the considered climate conditions and on the nature of the climate forcer. Considering an interactive stratospheric ozone chemistry under a $4xCO_2$ climate perturbation, Nowak et al. (2014) found a climate sensitivity of 1.05 $K.W^{-1}.m^{-2}$ in their ESM. Applying this climate sensitivity, the surface temperature change associated with the ozone feedback would be about 1.8 K when considering interactive stratospheric chemistry (compared with the EOCENE_Oz1855 run). The climate sensitivity to ozone change can obviously vary from one ESM to another since, for example, the sensitivity of the Brewer Dobson circulation to climate is highly model-dependant (SPARC CCMVal, 2010). Nonetheless, this estimation allows us to discuss the importance of considering this chemistry-climate feedback when attempting to simulate greenhouse paleoclimates. According to the IPCC AR5 report (IPCC, 2013), the global land surface air temperature anomaly is +12.7 K for the Early Eocene Climatic Optimum (Masson-Delmotte et al., 2013). This estimation is based on the simulations from several models analysed by Lunt et al. (2012) for which there was no common modelling protocol (e.g. $CO_2$ being in the PIx2 to PIx16 range). One of these models, the HadCM model, estimates that the effect of changing non-$CO_2$ boundary conditions (topography, bathymetry, solar constant and vegetation) for Eocene conditions leads to a 1.8 K increase in the global mean surface air temperature (to be compared to a 3.3 K increase when

doubling the $CO_2$). The feedback of stratospheric ozone on surface air temperature could thus represent about 15% of the total temperature anomaly reported between Eocene and preindustrial conditions and be as important as the effect of external forcings.

In strong greenhouse climate, the terrestrial carbon and nitrogen cycles are intensified, releasing high $CH_4$ and $N_2O$ in the atmosphere (Beerling et al. 2011). The effect of changing $N_2O$ and $CH_4$ in the troposphere (including the $H_2O$ increase in the stratosphere induced by the $CH_4$ increase) has been assessed by Beerling et al. (2011). These authors find, with the STOCHEM model, a 2.1K increase of global surface temperature due solely to the tropospheric composition changes. Our results suggest that the effect of the stratospheric ozone feedback on surface temperatures is of similar importance.

### 5.2 Impact on tropospheric conditions

Using an ESM including chemistry, Unger & Yue (2013) found that under warm and high methane Pliocene conditions, the stratospheric ozone burden was 5% higher than the preindustrial one. This stratospheric ozone increase resulted in a 20% reduction in the tropospheric photolysis rate of ozone ($O_3 + hv \rightarrow O^1D + O_2$) that leads to the formation of OH, the hydroxyl radical. This radical is the main oxidant of the troposphere and its decrease (of 20 to 25% in the Unger and Yue simulations) impacts the lifetime of chemical species and in particular $CH_4$. Our simulations show a 7.2% increase of the stratospheric ozone burden when comparing the EOCENE and the PREIND simulation (both including interactive chemistry) and a 8.8% difference when comparing the EOCENE simulation to the EOCENE_Oz1855.

In addition, we estimate the change in surface UV radiations and ozone photolysis in the Eocene conditions. Using the Radiation transfer model Quick TUV Calculator with a Pseudo-spherical discrete ordinate 4 streams (http://cprm.acom.ucar.edu/Models/TUV/Interactive_TUV/), we estimate the effect of the stratospheric ozone increase on the ozone photolysis, which controls the OH production. Considering a 65 DU change at a 50° latitude (corresponding to the maximum of Fig. 10), the photolysis rate at the surface decreases by 25%. A decrease in the ozone photolysis rate would induce a significant decrease in OH and hence in the tropospheric oxidizing capacity, thus making $CH_4$ longer lived and reinforcing its effect on climate. It would also impact the overall tropospheric chemistry.

## 6. Conclusion

The stratospheric dynamics and ozone layer respond to - and interact with - atmospheric variations (climate, tropospheric GHG content). In this study, we simulate these interactions in the case of the hot Eocene climate using a chemistry-climate model. We characterize the changes in ozone and middle atmospheric dynamics induced hot Eocene climate conditions characterized by a $4xCO_2$ climate, elevated concentrations of $CH_4$ and $N_2O$ and substantial changes in surface boundary conditions (e.g., sea surface temperature, sea-ice cover, topography, …) compared to preindustrial climate. The climate impact of the stratospheric response under hot conditions is also discussed.

Comparing the Eocene simulation with a preindustrial simulation, we find a sharp increase in ozone in the upper stratosphere (reaching 40% at 2-3hPa in the tropics) linked to the strong cooling of the stratosphere (up to -12 K at 10 hPa), which slows down the chemical destruction of ozone. Meanwhile, ozone is greatly reduced in the lower tropical stratosphere (up to 40%) due to the intensification of Brewer Dobson circulation. These results are in agreement with previous modelling studies that

considered current tropospheric composition and a 4x$CO_2$ climate change.

As a consequence of the opposite ozone changes in the tropics (enhanced ozone in the upper stratosphere, reduced ozone in the lower stratosphere), the tropical total column ozone (TCO) is not affected much by the difference in climate between the Eocene and preindustrial periods. On the contrary, at mid-latitudes and, to a lesser extent, in the polar regions, the TCO is considerably increased. The TCO meridional distribution is also strongly modified, exhibiting particularly pronounced mid-

latitudes maxima and steeper negative poleward gradient from these maxima. These changes in meridional distribution reflect significant polar vortex changes during the winter/early spring, especially in the Northern Hemisphere. The polar vortex becomes stronger and more extended equatorward under the Eocene conditions, thus reinforcing the isolation of the polar vortex air masses from the mid-latitudes in comparison with preindustrial conditions. In our simulations, the reinforcement of the stratospheric polar vortex under Eocene conditions and the acceleration of the stratospheric overturning

circulation (which seems contradictory at first) is consistent with a reduced intensity of the planetary waves activity and its interaction with the mean-flow in early winter, and, inversely, an strongly amplified wave activity and interaction with the mean flow in late winter compared to preindustrial conditions.

We then explore the possible role of the stratospheric ozone response in the establishment of the Eocene climate. For that purpose, we compare the simulations with interactive ozone with simulations forced by the use of preindustrial ozone

climatologies. The difference in global mean TCO between the Eocene simulation and simulations using preindustrial climatologies is two to three times higher than the change in ozone observed between 1960 and end of the 90s (the minimum TCO) and of the same order as the changes projected between 2000 and 2100. The ozone increase in the upper stratosphere in the case of Eocene interactive ozone warms the atmosphere by up to 3 K above 230 hPa. In the tropical lower stratosphere, zonal mean temperatures are up to 3.5 K lower for the Eocene stratospheric ozone compared to the preindustrial

ozone. These changes in the thermal structure of the middle atmosphere could, via atmospheric circulation teleconnections, have significant regional consequences. Using the sensitivity of surface temperatures to stratospheric ozone changes determined by Nowak et al. 2014 (though climate sensitivity varies among climate-chemistry models), we estimate the contribution of stratospheric ozone feedback to surface temperature change in Eocene hot climate simulations. We find that it is potentially as important as the effects of non-$CO_2$ boundary conditions (topography, bathymetry, solar constant &

vegetation) or uncertainties due to gaseous tropospheric chemistry. The results suggest that future studies exploring long-standing Cenozoic warm climates questions, such as the varying latitudinal temperature gradient during hothouse periods, would benefit from exploring and integrating -even if computing-time-costly- the feedbacks of ozone on atmospheric temperatures, rather than prescribing preindustrial values.

*Author contributions*. The idea of the study, the design of numerical simulations and the radiative forcing analysis come from S. Szopa. R. Thiéblemont performed all the analysis on atmospheric dynamics. S. Botsyun provided the Eocene boundary conditions. S. Szopa & R. Thiéblemont prepared the first draft of the manuscript. All co-authors contributed to its edition.

*Acknowledgements*. This work is supported by a grant from the French National Research Agency (ANR-16-CE31-0010 for the PALEOx project). This work was granted access to the HPC resources of TGCC under the allocation 2017-A0050102212 made by GENCI (Grand Equipement National de Calcul Intensif). The authors are thankful to the NCAR Atmospheric Chemistry Division (ACD) for the distribution of the NCAR/ACD TUV: Tropospheric Ultraviolet & Visible Radiation Model (URL: http://cprm.acd.ucar.edu/Models/TUV/) and the availability of their quicktool. We thank Y.

Donnadieu for preparing the paleogeographical conditions for Eocene simulations, and Marion Marchand for her advice in the use of the REPROBUS model and Jean-Louis Dufresne for helpful discussion on radiative forcing.

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

**Figures**

**Table 1: Set-up of LMDz**

| Setup Name | Ozone | $CO_2$ | $CH_4$ | $N_2O^*$ | SST |
|---|---|---|---|---|---|
| PREIND | Interactive | 285 ppm | | | AMIP |
| EOCENE | Interactive | 1120 ppm | 791 ppb | 275 ppb | Extracted from FOAM Eocene simulation |
| EOCENE_OzRoyer | Prescribed from Royer | | | | |
| EOCENE_Oz1855 | Prescribed from Szopa et al. (2013) | | | | |

Note: For the Eocene simulations, the REPROBUS chemical model consider $CH_4$ concentration of 3614 ppb and $N_2O$ concentration of 323 ppb.

**Table 2: Global change of total column ozone, temperature and effective radiative forcings induced by the use of an interactive stratospheric chemistry instead of climatologies.**

| | Interactive $O_3$ vs Royer (EOCENE- EOCENE_OzRoyer) | Interactive $O_3$ vs a 11 year mean climatology centered on 1855 (EOCENE- EOCENE_Oz1855) |
|---|---|---|
| Change of globally averaged TCO (DU) | 45.5 | 34.2 |
| Effective radiative forcing (W.m$^{-2}$) | 1.4 | 1.7 |
| Global Temperature change (K) | 0.4 | 0.3 |
| Stratospheric temperature change (K) above 230 hPa | 1.4 | 1. |

10 **Table 3. Differences in shortwave and longwave radiative fluxes at different vertical levels between the EOCENE simulation and the climate-only simulation with the 1855 ozone climatology.**

Net = 1.69 W.m$^{-2}$

| Downward | Upward | Net | Net SW+LW |
|---|---|---|---|
| | | | |

| | | | | | |
|---|---|---|---|---|---|
| Top of Atmosphere | Shortwave | 0.00 | -1.00 | 1.00 | 1.80 |
| | Longwave | | -0.80 | 0.80 | |
| 200hPa | Shortwave | -0.20 | -0.73 | 0.53 | 1.94 |
| | Longwave | -0.11 | -1.30 | 1.19 | |
| Surface | Shortwave | 0.30 | | | |

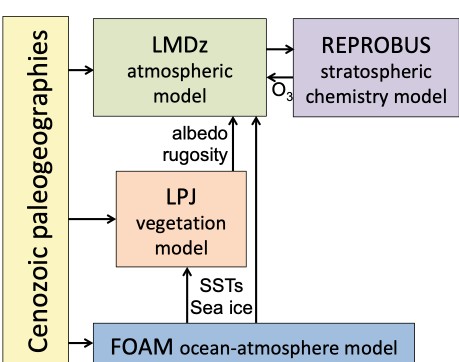

5    **Figure 1. Modelling set-up for the Eocene simulations**

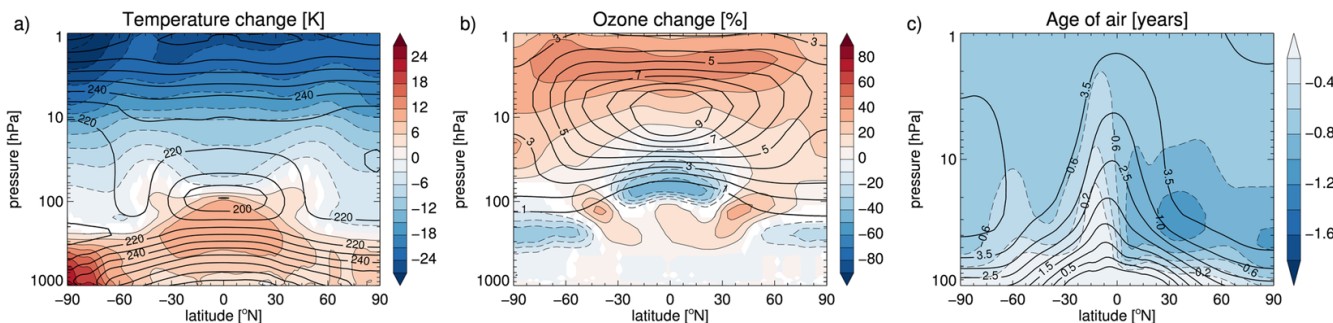

**Figure 2: Annual mean differences (EOCENE minus PREIND) of zonally averaged temperature (in K, panel a), ozone (in %, panel b) and age of air (in years, panel c). Color filled contours in a) and b) indicate that anomalies are**
10   **statistically different at the 1% confidence level according to a t-test. Black contours show the preindustrial climatology expressed in K (panel a), ppm (panel b) and years (panel c).**

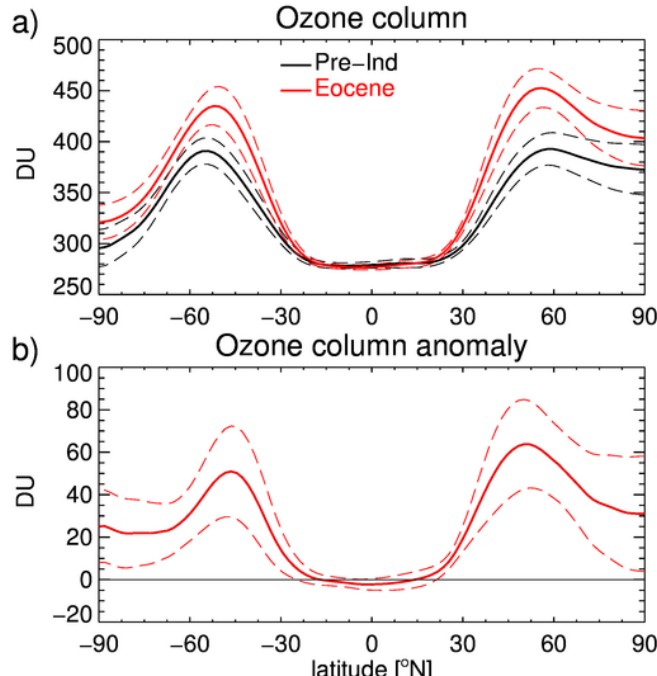

**Figure 3: Latitudinal profile of the total column ozone (in Dobson Unit or DU, panel a) in the (red) EOCENE and (black) PREIND simulation. Total column ozone change (in DU, panel b) between the EOCENE and PREIND simulation. Dashed lines delimit the 2-σ uncertainty envelop, which is represented by the standard error of the mean.**

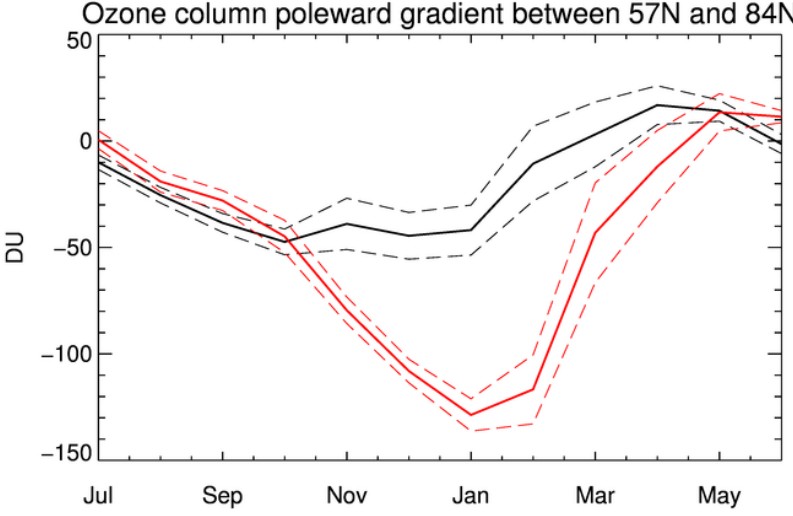

**Figure 4: Zonally averaged seasonal evolution of the latitudinal gradient (computed as the difference between 84° N and 57° N) of the total column ozone for the EOCENE (red) and PREIND (black) simulations. Dashed lines delimit the 2-σ uncertainty envelope, which is represented by the standard error of the mean.**

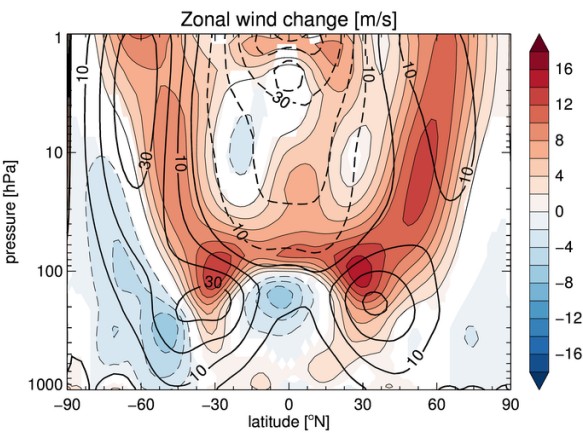

**Figure 5: Annual mean differences (EOCENE minus PREIND) of zonally averaged zonal wind (in m/s). Color filled contours indicate anomalies that are statistically different at the 1% confidence level according to a t-test. Black contours show the preindustrial climatology.**

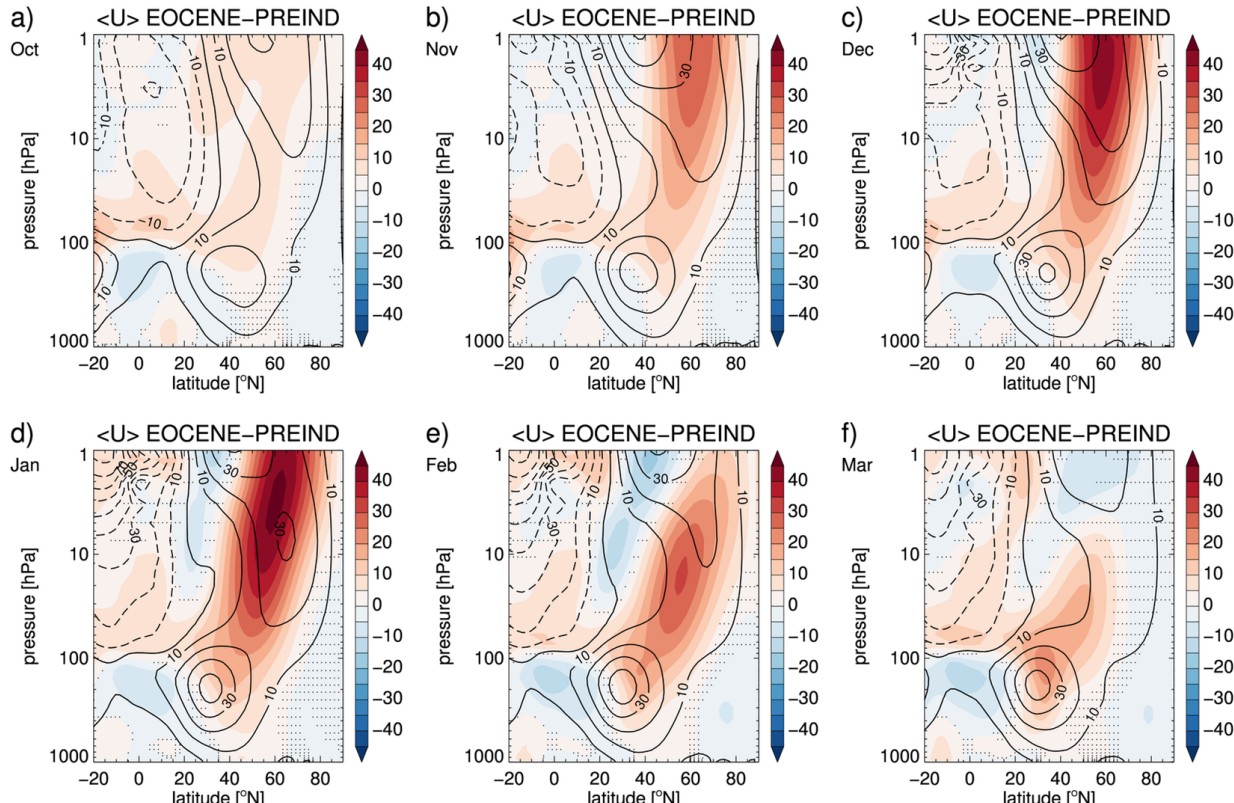

**Figure 6. Monthly evolution (October to March) of the zonal-mean zonal wind differences between the Eocene and preindustrial conditions in the Northern Hemisphere. Dotted regions indicate that anomalies are insignificantly different at the 5% confidence level according to a t-test. Black isolines shows the climatology derived from the preindustrial experiment.**

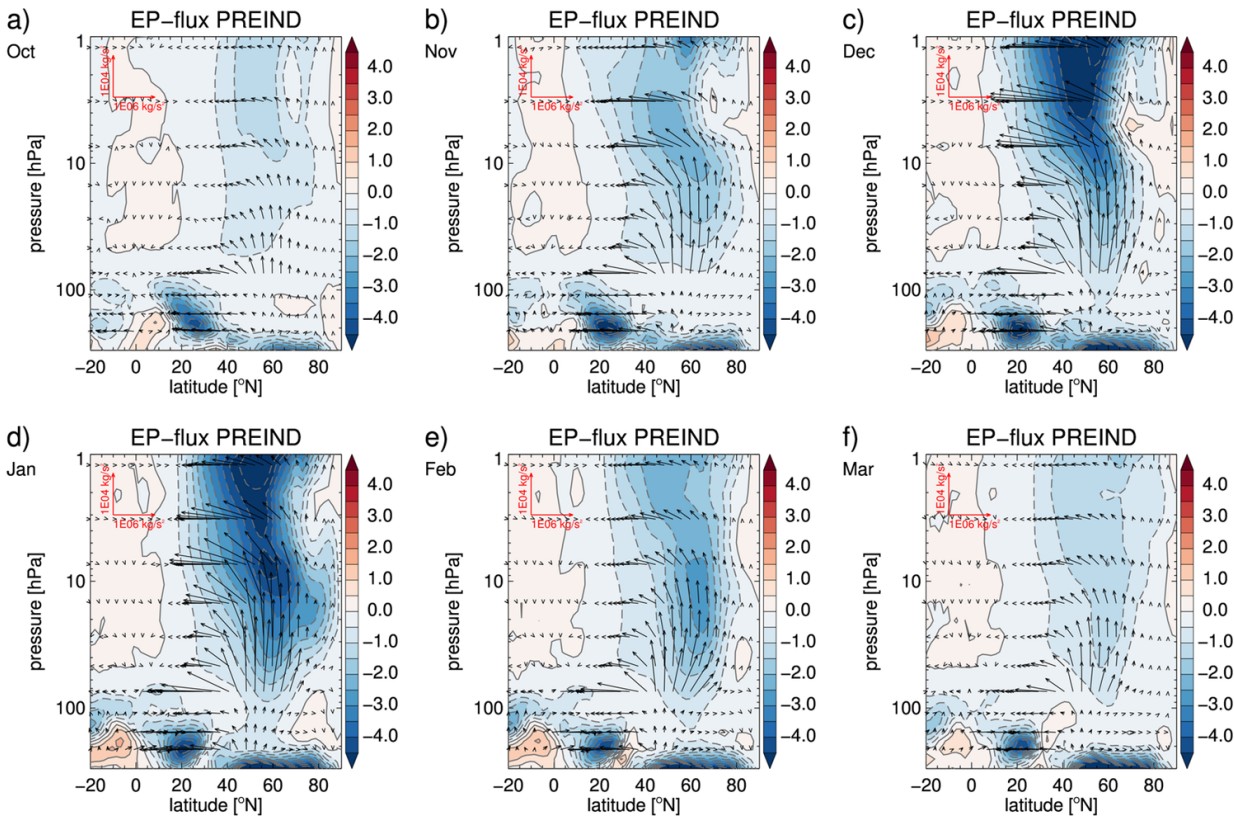

**Figure 7. Monthly evolution (October to March) of the Eliassen-Palm flux (vectors) and its divergence (contours, in m/s/d) under preindustrial conditions in the Northern Hemisphere.**

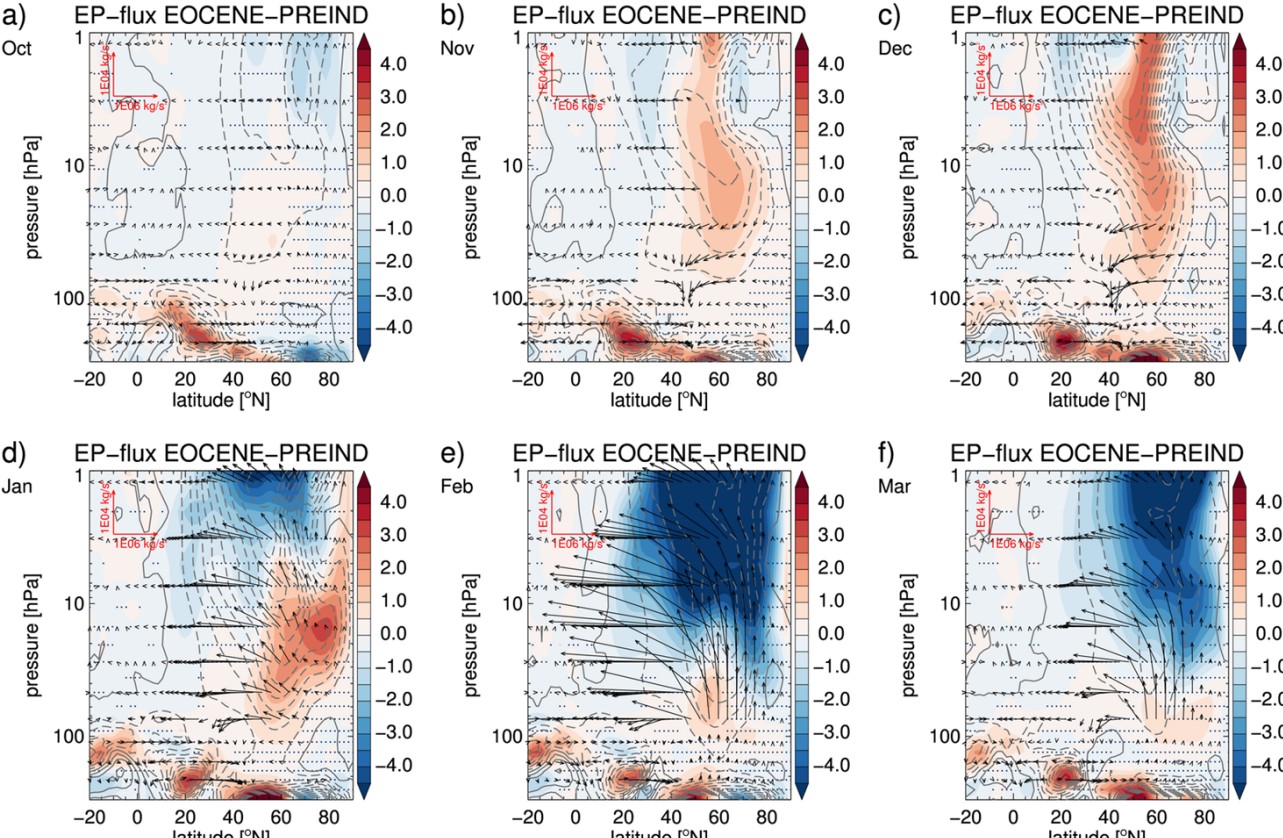

**Figure 8. Monthly evolution (October to March) of the differences between the Eocene and preindustrial conditions of the Eliassen-Palm Flux and its divergence. Dotted regions indicate that anomalies are insignificantly different at the 5% confidence level according to a t-test. Preindustrial climatology is shown with dashed contours.**

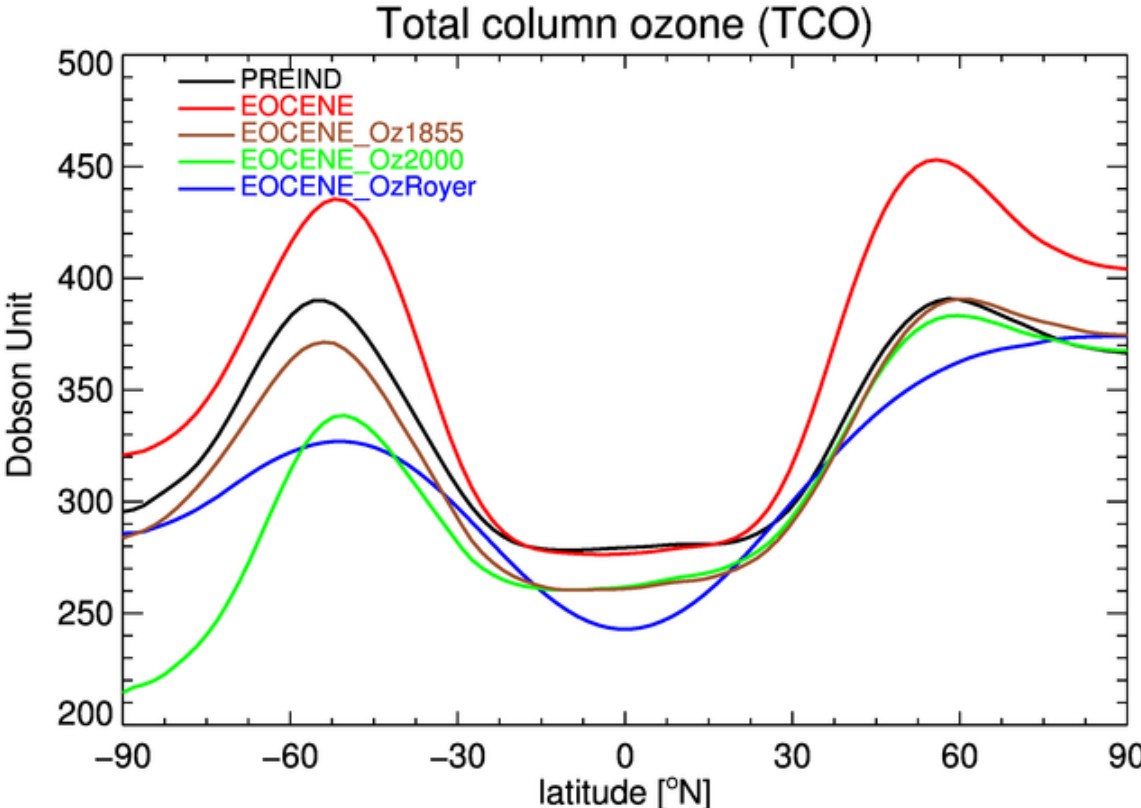

**Figure 9: Latitudinal distribution of ozone considered by the circulation model LMDz when using climatologies from Royer (blue), from Szopa et al. (2013) centered on the year 2000 (green) or centered on the year 1855 (maroon), or interactively computed by REPROBUS for Eocene conditions (red) or preindustrial conditions (black).**

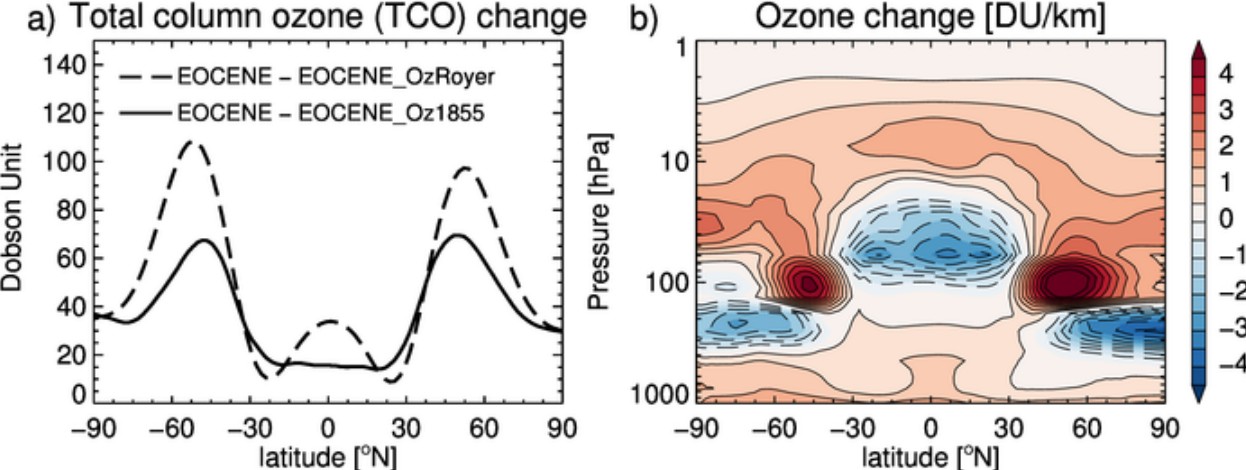

**Figure 10: Total column ozone (TCO) changes (in Dobson Units or DU) between the EOCENE simulation and the EOCENE-Oz1855 simulation (considering the 1855 ozone climatology) (panel a), and TCO changes between the EOCENE simulation and the two climate-only simulations considering the (solid) 1855 ozone climatology and (dashed) Royer ozone parameterization (panel b).**

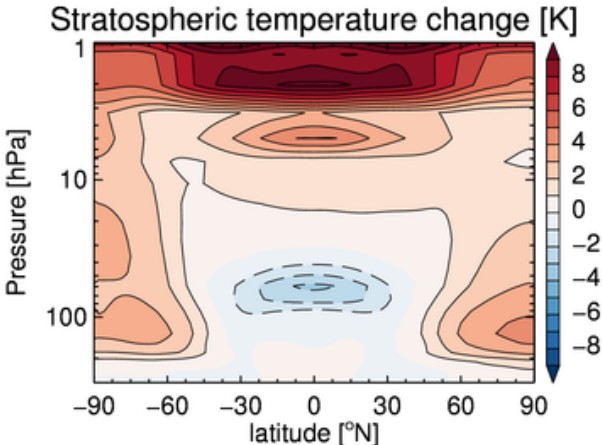

**Figure 11: Stratospheric temperature changes (K) between the EOCENE simulation and the climate-only simulation with the 1855 ozone climatology.**

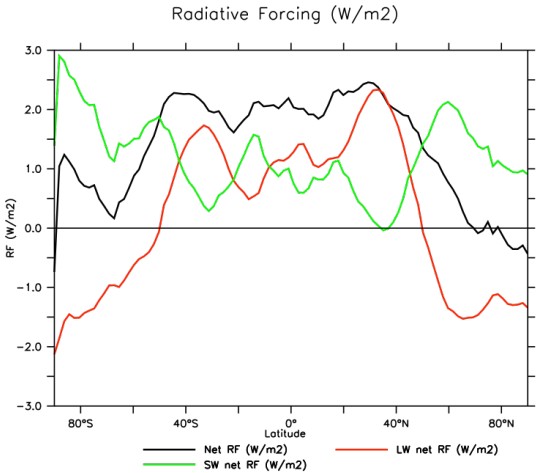

**Figure 12. Differences in radiative fluxes as a function of latitude** between the EOCENE simulation and the climate-only simulation with the 1855 ozone climatology

