# Peer review of "Role of the stratospheric chemistry-climate interactions in the hot"

_Climate of the Past, 2018_

## Referee Comment (RC1) · Anonymous Referee #1 · 6 Nov 2018

**General comment**

The manuscript presents the results of the ozone layer simulations during Eocene and preindustrial conditions as well as comparison with several climatological ozone datasets. The authors applied several models including Atmosphere-ocean GCM and vegetation model driven by the Eocene boundary conditions as well as chemistry-climate model LMDz-Reprobus (CCM LR). The authors analyze the ozone layer response to the enhanced concentrations of $CO_2$, $CH_4$ and $N_2O$ comparing with preindustrial run. The obtained results are mostly known from several previously published estimates of the ozone response to climate warming. The differences with the published results consist of very strong acceleration of the Northern polar night jet resulting in smaller total column ozone increase over the high-latitudes. The comparison with

other climatological datasets for preindustrial and present day conditions shows expected differences. The authors interpret this results as a necessity to use interactive ozone chemistry for the simulation of the Eocene or other extremely warm climates. This conclusion is supported by the strong (about 1.8 W/m**2) radiative forcing caused by switch from prescribed to interactive ozone. The obtained results are potentially interesting, but a lot of additional explanations and clarifications are needed before I can recommend the publication.

Major issues

1. Section 2: The used model setup is not clearly presented. I understand that FOAM+LPJ models are used to produce boundary conditions (topography, geography, 4xCO2 and so on) for Eocene. Then, in the section 2.2.2 it is said that CCM LR uses SST and land surface conditions. Does CCM LR use proper topography and land configuration? If the simulated period is around 55 Ma, why nothing is said about oxygen mixing ratio which was only about 17% and large increase of biogenic emissions (i.e., isoprene). These components can substantially change ozone mixing ratio in the both stratosphere and troposphere. How successful is FOAM simulations of the past climate? As far as I know the Eocene climate with no substantial horizontal temperature gradients is difficult to reproduce. From the first paragraph on page 6 I understood that CH4 and N2O have not been included in the CCM LR radiation code, but their influence is implicitly included in CO2. How exactly it was done? Did the author use greenhouse warming potential or some other scaling technique? I suggest rewriting section 2.2 to make it more understandable.

2. Section 3: Most of the results of this section agree well with several previous publications. On unexpected results is strong acceleration of the boreal polar night jet, which is more than two times stronger during Eocene. The authors explain it by the extra cooling of polar cap area by enhanced CO2. This result does not agree with previous publications. For example, for 4xCO2 case the acceleration of zonal wind was not detected (e.g., Ferraro et al., 2015, doi:10.1002/2014JD022734, Figure 4). Theoretically, it should be expected because the enhanced CO2 cools down stratosphere everywhere and does not build up additional horizontal gradients. Maybe the cause is not CO2? It should be clarified and analyzed. I would also suggest shortening second paragraph on page 8. I guess, most of the readers know the basic atmospheric dynamics.

3. Section 4: First of all, the considered effects are not related to interactive chemistry but rather to the use of not appropriate ozone field. I guess, most of the differences discussed in this section disappear if the authors use the model w/o interactive chemistry, but with the ozone field prescribed from the Eocene run. I do not see any reason to compare Eocene run with the results of the model run driven by OzRoyer. Obviously, there will be substantial difference due to different situation during Eocene and present day. Comparison with Oz1855 is also not instructive because the ozone field is very close to the results of preindustrial run.

4. Section 5: The problem here is related to the magnitude of radiative forcing. 1.8 W/m**2 from stratospheric ozone increase looks extremely overestimated and has probably wrong sign. Forster et al., 2011 (doi: 10.1029/2010JD015361) showed using very accurate LBL radiation codes that 10% decrease of the stratospheric ozone gives only about 0.25 W/m**2 (their Table 4). Portman and Solomon (2007, doi:10.1029/2006GL028252) concluded that the ozone radiative forcing caused by warming climate is within 0.1 W/m**2. I think that very large 1.8 W/m**2 ozone forcing (comparable to anthropogenic radiative forcing during 21st century) should be clearly explained. At least, its geographical, vertical and spectral signatures should be illustrated. In my opinion this forcing can be generated only by extraordinary high increase of the tropospheric or UTLS ozone (e.g., Beerling, 2011), which is not visible from presented results. The estimation of the surface temperature response using some other model sensitivity to homogeneous radiative forcing is oversimplified. If the obtained 1.8 W/m**2 is true (which I doubt) it will show the importance of the problem by itself.

---

## Short Comment (SC1) · 12 Nov 2018

I am a MSc student at Uppsala University with an interest in past climate and past climate research. The potential role of stratospheric ozone chemistry was evaluated in this study with a stratospheric chemical-climate model in the case of the Eocene hot conditions (4xCO2 climate and high concentrations of tropospheric N2O and CH4). Their results show that the ozone layer is significantly different under those conditions, with enhanced ozone column at mid-high latitudes and more or less unchanged on tropical latitudes. Their result suggests that using calculated stratospheric ozone by the model (instead of preindustrial ozone distribution) can change the global air temperature by 14% and highlights the sensitivity of ozone to hot climate conditions and the chemical composition of the atmosphere. Their result is significant since the cli-

mate sensitivity to stratospheric ozone feedback differs largely between models and need to be constrained better for both modelling of past climates as well as futures climates.

The aim and purpose of the study is clearly defined, and why the subject is important in this day and age are clearly formulated. Necessary information about atmospheric chemistry and the conditions in the different parts of the atmosphere, as well as the modelling are provided with relevant studies and references. The introduction gives good information on atmospheric chemistry interactions, and what studies haves been done before including their limitations. The method (the modelling) to address their aim is well stated and described. The parameters and limitations are also thoroughly described with relevant references included. The table of the settings for the simulations could however be good to have connected to the text to enable the reading flow. Their findings are clearly coupled with their method with high transparency (uncertainties, assumptions etc.). Their interpretations seem logical and reasonable, and coupled to the relevant atmospheric conditions and feedbacks. It would have been easier to follow their results and interpretations if accompanied by tables and figures, instead of having them in an appendix, where you have to go back and forth to get the whole picture of the results, interpretations and how they differed from the different simulations. The figures themselves are easy to read with the uncertainties clearly outlined. The arguments for why stratospheric ozone and distributions are important in modelling for past and future climates are well formulated and stated in their discussion, coupled with relevant similar studies and references. I would like to see more references in the conclusion, especially when the results are compared with other studies to support their conclusions.

The article is well-structured and with clear separation of the different sections with a distinct and logical link between them. These sections are also very detailed. It is easy to follow how the team has performed their study and what components has been taken in count, and why those is important and how they affect each other.

The language used is appropriate and scientific without being hard to follow. Some sentences could be very long, which made it a bit harder to follow the reasoning. My suggestion is to see if some sentences could be shortened or divided into two or more sentences.

Over all a very detailed study poorly understood component of the atmosphere linked to climate and climate feedbacks. The potential role of stratospheric ozone chemistry and distribution show indications of having a significantly impacts on the climate, especially regarding feedback mechanisms, and as suggested should be constrained better in models to further investigate it's importance and affects. Especially, since its often neglected in today's climate models. I think this was a good quantitative study on the ozone layer's role and dynamic with climate, with a clear structure, necessary complementary information and logical and reasonable interpretation of their result. With some minor corrections the transparency and reasoning could be even easier to follow, e.g. to place the tables and figures with the section of text where they are referred instead of in an appendix.

---

## Referee Comment (RC2) · Anonymous Referee #2 · 3 Jan 2019

The role of stratospheric ozone changes in Eocene climate conditions is studied using coupled chemistry-climate simulations. These hot climate conditions are simulated by setting CO2 to 4 time its preindustrial (PI) control value, as well as setting tropospheric CH4 and N2O to significantly higher values compared to PI control. Much of the qualitative atmospheric temperature and ozone response resembles that of 4xCO2 experiments, which are well documented in the literature: tropospheric warming and stratospheric cooling, increased upper stratospheric ozone and decrease ozone in the lowermost tropical stratosphere indicative of an accelerated Brewer-Dobson circulation (BDC). Radiative feedbacks due to the modified ozone distribution are then argued to potentially play an important role in Eocene-like climates, of similar importance as non-CO2 boundary conditions.

[Figure]

Climate feedbacks due to stratospheric ozone have recently received some attention with different studies coming to different conclusions - compare Dietmueller et al. (2014), Nowack et al. (2015), Marsh et al. (2016) - e.g., summarized in Chiodo et al. (2018). This study could provide a valuable contribution to this line of research and extend it to past climates. However, as presented I see a number of issues that need to be dealt with before publication. Perhaps this is possible within a major revision, but it may require a more substantial effort.

Major issues

The main motivation appears to be the climate sensitivity to stratospheric ozone, but this cannot be studied with the model setup used: because SSTs and other lower boundary conditions are prescribed, surface climate cannot respond to atmospheric changes. In fact, the prescribed SSTs, which come from a low-resolution coupled climate model (FOAM, which as far as I understand does not include interactive chemistry), are likely inconsistent with the ozone feedback that would result from the interactive chemistry simulations with LMDz.

Given this issue, another motivation is to study the stratospheric response (ozone, circulation) to the prescribed Eocene conditions. This would be fine, but in the current presentation in the manuscript this appears to be poorly conceived:

a) Only annual mean cross sections are shown, even though the essential dynamics take place in each hemisphere's winter and spring seasons. I think that seasonal-mean plots are necessary to substantiate the results and interpretations related to the stratospheric response.

b) Changes to tropospheric CH4 and N2O are applied but no results are presented that show what the effects of these changes are (N2O should lead to modified chemistry, as noted by authors; CH4 could lead to changes in stratospheric H2O). If these changes are deemed to not be so important then why not simply perform simulations with 4xCO2, which would also have the advantage of providing better comparability

to previous results with similar forcing? If above changes of tropospheric species are deemed to be important then this calls for corresponding analyses and results to be presented.

c) How well does this model (LMDz) simulate the stratospheric circulation compared to other state-of-the-art chemistry-climate models? The CCMVal-2 activity included a version of this model, which indicates it is performing well in terms of several diagnostics, but also has some issues (see SPARC CCMVal report referenced): e.g., huge warm bias in upper stratosphere, where the radiative scheme seems to behave questionably, bias in surface energy balance, large cold bias in SH leading to strongest ozone hole out of all compared models etc. The only place in the literature where I could find a plot of the model's overturning streamfunction (i.e., its BDC) is Dietmueller et al. 2018 (Fig. 1 therein): it looks completely off, questioning the model's ability to simulate stratospheric transport (despite the fact that its AoA distribution looks okay) . . . in all fairness, Dietmueller et al. note that this may be a diagnostic, rather than an actual model problem. In any case, the authors should include information about the basic model performance in regards to stratospheric dynamics, transport, and climate, and convince the reader that this is a suitable model for the purposes of the study.

d) Other studies on ozone changes due to 4xCO2 (see references listed above) have highlighted the crucial role of changes in stratospheric H2O, which come about due changes in tropical tropopause temperature, but also due to ozone-temperature feedbacks near the tropical tropopause. This type of sensitivity could be important in order to understand the climate response to 4xCO2 and should be included in the results and discussion.

e) At face value, the presented results indicating both an accelerating BDC and stronger polar vortex seem to contradict each other, since a stronger BDC should be associated with stronger wave drag, which would be consistent with a weaker vortex. This is not discussed in the paper but seems important to understand the stratospheric changes. My guess is that this can be explained by the seasonality in the changes (cf.

Figs. 5, 6): the wave forcing seems indeed weaker in early winter when the vortex is much stronger (and I would expect a weaker BDC during that part of the season, but this should be checked and potentially included in the presented results). During late winter and spring the wave forcing is much enhanced consistent with an accelerated BDC - again this should be checked based on residual circulation diagnostics.

f) It is claimed that the stratospheric cooling due to higher $CO_2$ levels explains the changes in polar vortex strength, but why would the $CO_2$ cooling affect the meridional temperature gradient rather than lead to a meridionally uniform cooling, which would not affect the polar vortex? With the presented results the cause of the strengthened vortex remains confusing.

Minor Comments:

Fig. 1: why not present AoA similar to panels a, b (difference as color shading with PI control as black contours)? Also: what are the units for the presented PI O3?

page 4, section 2.1: it would help to include some information about how the model compares to other chemistry-climate models (see major comment)

page 4, line 31: please also provide the model top

page 5, line 3: "snapshots" - do you mean "time slices"?

page 5, line 17: please explain "LPJ"

page 5, line 24: please provide justification / motivation for why you choose a $CO_2$ value at the low end of what's recommended

page 6, line 1-2: please provide more detailed explanation for why radiative effect due to enhanced $CH_4$ and $N_2O$ levels would be accounted for by enhanced $CO_2$?

page 6, line 10: "80s" - you mean the 1980's? Is this meant to represent an "ozone-hole climate"? Why not simply use the O3 field from, e.g., a CCMVal-2 chemistry-climate simulation with your model?

page 6, line 25: please discuss the temperature changes a bit more, e.g.: is the Antarctic amplification (largest temperature response over Antarctica) a well-known response for these types of simulations? Why is there no corresponding Arctic amplification as happens for current climate change and happens for pure 4xCO2 runs?

page 7, line 13: "total ozone column" - here and elsewhere: usually this is referred to as "total column ozone (TCO)" and I'd recommend nomenclature consistent with other literature

page 8, line 3, Fig. 4: you already showed an indication that the winter season matters most, so why not show DJF and JJA changes instead of the annual mean (see major comments)?

page 9, line 1: "... drives the strength of the zonal wind" - 1) thermal wind balance doesn't tell you about cause and effect, so "drives" is misleading, 2) it's a relation between the meridional temperature gradient and the vertical zonal wind gradient (not the wind itself), so you wouldn't necessarily expect the temperature gradient at 10 hPa to correspond to the wind at 10 hPa ...

page 9, line 2: the heat flux is a proxy for the vertical Eliassen-Palm ($\sim$wave activity) flux, which more accurately also involves the vertical temperature gradient and the background vorticity; given that you compare two very different climates, I wonder whether the heat flux is a sufficiently accurate measure of wave activity flux, since both background temperature and vorticity structures might contribute?

—————————————

---

## Author Comment (AC1) · 12 Feb 2019

We thank the **reviewer #1** for his/her comments which should lead to a clearer manuscript, notably regarding the simulation set-up, the understanding of processes leading to changes on polar night jet strength and the radiative forcing due to stratospheric ozone changes.

Hereafter, we explain how we are able to improve the paper regarding the issues mentioned by the reviewer #1 in blue italic (actions taken on the manuscript are preceded by an arrow).

*1. Section 2: The used model setup is not clearly presented. I understand that FOAM+LPJ models are used to produce boundary conditions (topography, geography, 4xCO2 and so on) for Eocene. Then, in the section 2.2.2 it is said that CCM LR uses SST and land surface conditions. Does CCM LR use proper topography and land configuration?*

The FOAM+LPJ model is prescribed with a 4x $pCO_2$ and an Eocene paleogeography to compute sea-surface temperatures, sea-ice extent and vegetation surface properties (albedo, roughness, landcover). We use these surface properties as boundary conditions for LMDz-REPROBUS (not CCM LR), together with the same Eocene paleogeograhy and greenhouse gas concentrations.

=> we have clarified the description of this set-up in the text and add a sketch describing the modelling set-up.

*If the simulated period is around 55 Ma, why nothing is said about oxygen mixing ratio which was only about 17% and large increase of biogenic emissions (i.e., isoprene). These components can substantially change ozone mixing ratio in the both stratosphere and troposphere.*

Oxygen variations are poorly constrained over such timescales. There is no consensus on the oxygen variations through the Cenozoic (see Fig 1 of Wade et al. 2018 presenting the different oxygen content reconstructions in the Phanerozoic and discussing the methodologies used to produce them in their introduction https://www.clim-past-discuss.net/cp-2018-149/cp-2018-149.pdf). In view of these uncertainties, the climate models use a present-day oxygen content to investigate past climates except studies specifically investigating the potential role of oxygen variations but in this case they focus on more ancient periods for which the estimations of large oxygen variations are in better agreement (e.g. Wade et al. 2018, Charnay et al. 2013).

**Ref:**

Wade, D. C., Abraham, N. L., Farnsworth, A., Valdes, P. J., Bragg, F., and Archibald, A. T.: Simulating the Climate Response to Atmospheric Oxygen Variability in the Phanerozoic, Clim. Past Discuss., https://doi.org/10.5194/cp-2018-149, in review, 2018.

Charnay, B., Forget, F., Wordsworth, R., Leconte, J., Millour, E., Codron, F., and Spiga, A.: Exploring the faint young Sun problem and the possible climates of the Archean Earth with a 3-D GCM, Journal of Geophysical Research: Atmospheres, 118, 10,414–10,431, https://doi.org/10.1002/jgrd.50808, 2013.

=> In the revised version, we now explain briefly the choice we made for oxygen content.

The suspected large increase in biogenic VOC (due to warmer climate) is of major concern for tropospheric chemistry but BVOC do not reach the stratosphere in significant amounts, being very quickly oxidized in the troposphere. However, they can impact the methane concentrations by altering the tropospheric oxidizing capacity as found by Beerling et al. [2011]. That's why we use, in our simulations, the $CH_4$ concentrations calculated by Beerling et al. [2011] who studied the tropospheric chemistry under Eocene conditions using a land-tropospheric chemistry-climate model.

*How successful is FOAM simulations of the past climate? As far as I know the Eocene climate with no substantial horizontal temperature gradients is difficult to reproduce.*

FOAM alone has been used specifically for the Eocene period in previous published studies (e.g., Zhang et al., 2012). In addition, numerous recently published paleoclimate studies are based on the two-step methodology based on FOAM and LMDZ and this set-up has been shown to perform well [e.g. Botsyun et al. 2019, Ladant et al., 2014; Ladant et al. 2016; Licht et al., 2014; Pohl et al. 2016; Porada et al. 2016].

Moreover, even if climate models, including FOAM, exhibit weaknesses in simulating past climates, this does not prevent exploring middle atmospheric ozone dynamic for paleo-climate modelling, in particular, under warm climates. One has to keep in mind that the main objective of the paper is to study stratospheric ozone changes, which have been neglected so far, under Eocene conditions and their potential importance for climate.

=> We now provide more information and references for LMDz-FOAM in the set-up section.

**Ref:**

Botsyun S., P. Sepulchre, Y. Donnadieu, C. Risi, A. Licht, J. K. Caves Rugenstein. "Revised paleoaltimetry data show low Tibetan plateau elevation during the Eocene.", Science (2019), *in press*.

Ladant, J. B., Y. Donnadieu, V. Lefebvre, and C. Dumas (2014), The respective role of atmospheric carbon dioxide and orbital parameters on ice sheet evolution at the Eocene-Oligocene transition, Paleoceanography, 29, 810–823, doi:10.1002/2013PA002593.

Ladant, J.-B. & Donnadieu, Y. Palaeogeographic regulation of glacial events during the Cretaceous supergreenhouse. Nat. Commun. 7:12771, doi: 10.1038/ncomms12771 (2016).

Licht, A., et al. (2014), Asian monsoons in a late Eocene greenhouse world, Nature, 513(7519), 501–506.

Pohl, A., Y. Donnadieu, G. Le Hir ,J.-B. Ladant, C. Dumas, J. Alvarez-Solas, and T. R. A. Vandenbroucke (2016), Glacial onset predated Late Ordovician climate cooling,Paleoceanography,31, 800–821,doi:10.1002/2016PA002928.

Porada, P. et al. High potential for weathering and climate effects of non-vascular vegetation in the Late Ordovician. Nat. Commun. 7:12113 doi: 10.1038/ncomms12113 (2016).

Z Zhang, F Flatøy, H Wang, I Bethke, M Bentsen, Z Guo, Early Eocene Asian climate dominated by desert and steppe with limited monsoons, Journal of Asian Earth Sciences, Volume 44, 2012, https://doi.org/10.1016/j.jseaes.2011.05.013.

*From the first paragraph on page 6 I understood that CH4 and N2O have not been included in the CCM LR radiation code, but their influence is implicitly included in CO2. How exactly it was done? Did the author use greenhouse warming potential or some other scaling technique? I suggest rewriting section 2.2 to make it more understandable.*

We agree that this point was not well explained.

Even if inferred from proxies, the temperature changes for Eocene are better known than the greenhouse gases content (for which only $CO_2$ level can be inferred but with large uncertainties). For this reason, paleoclimate modellers have investigated the Eocene climate running simulations with various $CO_2$ covering a large range of concentrations with the aim of reproducing the amplitude of surface temperature changes. They do not change the level of other GHGs because no data are available on them. However, as only $CO_2$ is adjusted to match the temperature difference it means that the $CO_2$ perturbation implicitly represents the sum of all the GHG perturbations. This methodology is the one recommended by Lunt et al. 2017 for the DeepMIP project. For that reason we only perturb $CO_2$ in FOAM and LMDz for Eocene simulations.

Thus, in the LMDz-REPROBUS chemistry-climate modelling, fixed $CO_2$, $CH_4$ and $N_2O$ are used as inputs to the radiative scheme. As a result, only ozone changes influence the climate during a preindustrial or Eocene simulation. That way, the effect of ozone changes on middle atmospheric climate can be isolated and quantified. Obviously, ozone chemistry is also affected by changes in $N_2O$ and $CH_4$, two

key stratospheric source gases. To account for this effect, there are $CH_4$ and $N_2O$ chemically active tracers (i.e. modified by the transport and chemistry schemes) and whose surface concentrations are taken from the modelling study of Beerling et al. 2011. Their global distributions change with time during a simulation, but they are not used as inputs to the radiative scheme and hence their changes do not affect the climate, only ozone changes do.

=> we have now clarified this point in section 2. 2 and in the Table 1

*2. Section 3: Most of the results of this section agree well with several previous publications. On unexpected results is strong acceleration of the boreal polar night jet, which is more than two times stronger during Eocene. The authors explain it by the extra cooling of polar cap area by enhanced CO2. This result does not agree with previous publications. For example, for 4xCO2 case the acceleration of zonal wind was not detected (e.g., Ferraro et al., 2015, doi:10.1002/2014JD022734, Figure 4). Theoretically, it should be expected because the enhanced CO2 cools down stratosphere everywhere and does not build up additional horizontal gradients. Maybe the cause is not CO2? It should be clarified and analyzed.*

Indeed, the $CO_2$ cooling of the stratosphere should be uniform and hence would not affect the meridional temperature gradient. We thank the reviewer for noticing this flawed explanation, which is now removed from the revised version. This comment has prompted us into analyzing more thoroughly the mechanism. As shown on Figure 2b of the manuscript, however, the increase of ozone in the upper stratosphere could inflect further the shortwave heating rates gradient in the winter hemisphere. This is indeed what we found with an overall more negative meridional SW heating gradient throughout the depth of the stratosphere, which maximizes at the stratopause. This effect is however balanced by the fact that equatorial temperatures are decreasing at a faster rate than the polar temperatures in the Eocene simulation. So the net difference is slightly positive in the middle stratosphere and negative at the stratopause level. Radiative effects on the polar vortex in early winter due to $CO_2$ increase appear hence to be very modest compared to the changes in the wave activity as we describe further in the following.

As shown on Rev1 Figure 1 below, a much stronger stratospheric polar vortex develops in early winter (Nov-December) under the Eocene conditions compared to pre-industrial conditions. The strength of the polar night jet is doubled over the full depth of the stratosphere. By late winter (starting in January), the anomalies progressively reverse from the upper part of the stratosphere. In March, the stronger polar vortex anomalies in the middle atmosphere are no longer significant. To understand better such a seasonal evolution of the polar vortex, we have performed some transformed eulerian mean calculations [*Andrews et al.*, 1987] in order to diagnose the resolved wave activity changes and their interaction with the mean flow. We first examine the climatology of the EP-flux and its divergence in the preindustrial experiment and then analyse the anomalies when moving to Eocene conditions.

[Figure]

**Rev1 Figure 1**. Seasonal evolution (Oct to Mar) of the zonal mean zonal wind differences between the Eocene and preindustrial conditions. Shaded contours indicate that anomalies are significant at the 5% levels according to a t-test. Black contours show the preindustrial run climatology.

The preindustrial climatology of the planetary wave propagation (EP-flux) and its interaction with the mean flow (EP-Flux divergence) shows that, permanently in winter, the wave activity penetrates in the stratosphere and the breaking of planetary waves leads to westward momentum forcing which maximizes near the location of the southern flank of the polar night jet (Rev1 Figure 2). This contributes to erode and weaken the polar vortex, to a warming of the polar stratosphere and lead to a net poleward residual mass transport (which contributes the Brewer-Dobson circulation to a large extent). The wave activity and its interaction with the mean flow peaks in December-January but is already large in November (which can eventually lead to SSW [e.g. de la Camara et al., 2016]).

[Figure]

**Rev1 Figure 2**. Seasonal evolution (Oct to Mar) of the Eliassen-Palm Flux (vectors) and its divergence (contours, in m/s/d) under preindustrial conditions.

Under Eocene conditions (Rev1 Figure 3), it appears that the planetary wave penetrating the stratosphere in early winter (Nov-Dec) is significantly reduced and deflected equatorward as revealed by the EP-Flux vector pointing downward and equatorward in the mid-latitude lower stratosphere (see also Figure 7b showing a lower eddy heat flux at 100 hPa in the former version of the manuscript). This is associated with an anomalous positive E-P flux divergence throughout the depth of the stratospheric polar night jet (near 60°N), which indicates a reduced westward momentum forcing by waves and hence allows a stronger development of the polar vortex in early winter. In contrast, by January, the wave activity becomes significantly larger (see also Figure 7b), the westward forcing appears strongly amplified in the upper stratosphere and this momentum forcing anomaly progressively propagates downward. This is consistent with the reversal of the zonal-mean zonal wind anomaly in the upper stratosphere, but also with the overall extremely rapid deceleration of the polar vortex strength seen on Figure 6 of the former version of the manuscript. Relatively, the Brewer-Dobson circulation will hence be less reduced in early winter than accelerated in late winter (where the differences in the wave forcing are much stronger), resulting in a net acceleration of the Brewer-Dobson under Eocene conditions compared to preindustrial conditions as revealed by the younger age of air. Note that we also examined the residual circulation velocities v* and w* which confirms the seasonal changes in the Brewer-Dobson circulation strength (not shown).

[Figure]

**Rev1 Figure 3**. Seasonal evolution (Oct to Mar) of the differences between the Eocene and preindustrial conditions of the Eliassen-Palm Flux and its divergence. Shaded contours indicate that anomalies are significant at the 5% levels according to a t-test. Preindustrial climatology is shown with dashed contours.

The subsequent question is what causes these seasonal changes in the wave activity and its forcing on the mean flow. Potential factors that may play a role in modulating the wave activity are for instance SST changes (e.g. Hu et al., 2014), sea-ice changes (e.g. Kim et al., 2014), tropospheric wind changes (e.g. Karpechko and Manzini, 2017), and topography (Shi et al., 2014). However, to better quantify the relative importance of the above-mentioned various factors, a thorough detection/attribution experimental protocol would be needed. Such a detection/attribution study goes beyond the scope of our study whose aim is to characterize the stratospheric background state under Eocene-like extreme climate conditions. Nonetheless, we could analyze additional Eocene and Preindustrial experiments that have been performed with the same atmospheric model (LMDz) but without interactive chemistry (i.e. preindustrial ozone is prescribed) and with a flat topography. It appeared very clearly that changes in the topography have first order effects on wave development and propagation and hence on the stratospheric polar vortex. Between the Eocene and the preindustrial eras, beside large changes in the topography (see Figure below), changes in air-sea thermal contrasts, sea-ice cover, sea surface temperature, can all have a substantial effect on the background state atmospheric circulation, wave activity and hence the stratosphere dynamics. The complexness of these effects and their possible interactions make an unambiguous attribution impossible in the absence of a devoted experimental protocol that is out of reach for the present study.

=> In the revised version of the manuscript, the seasonal stratosphere dynamics analysis shown above is now the section 3.2, which has been renamed "Seasonal evolution of the Northern Hemisphere stratospheric polar vortex in Eocene conditions". Figures 5 and 6 have been removed and replaced by the three Figures shown above. Figure 5 has been moved in the supplementary material as we believe that it illustrates very clearly the seasonal changes of the polar vortex. Topographic changes shown below have also been added in the supplementary material.

[Figure]

**Rev1 Figure 4**. Topography (in m) for (left) preindustrial and (right) early Eocene (-55 Ma) periods used as LMDz boundary conditions.

**Ref**

de la Cámara, A., Lott, F., and Abalos, M.:Climatology of the middle atmosphere in LMDz: Impact of source-related parameterizations of gravity wave drag, J. Adv. Model. Earth Sys., 8, 1507– 1525, https://doi.org/10.1002/2016MS000753, 2016.

Hu, D. Z., W. S. Tian, F. Xie, J. C. Shu, and S. Dhomse, 2014: Effects of meridional sea surface temperature changes on stratospheric temperature and circulation. Adv. Atmos. Sci.,31(4), 888–900, doi: 10.1007/s00376-013-3152

Karpechko, A. Y., & Manzini, E. (2017). Arctic stratosphere dynamical response to global warming. Journal of Climate, 30(17), 7071–7086. https://doi.org/10.1175/JCLI-D-16-0781.1

Kim, B.M., S.W. Son, S.K. Min, J.H. Jeong, S.J. Kim, X. Zhang, T. Shim, and J.H. Yoon (2014), Weakening of the stratospheric polar vortex by Arctic sea-ice loss, Nat. Commun., 5, doi:10.1038/ncomms5646.

Shi, Z., Liu, X., Liu, Y., Sha, Y., Xu, T., 2014. Impact of Mongolian Plateau versus Tibetan Plateau on the westerly jet over North Pacific Ocean. Climate Dynamics 44 (11-12), 3067–3076. 7

*I would also suggest shortening second paragraph on page 8. I guess, most of the readers know the basic atmospheric dynamics.*

The paragraph is now removed from the revised version. Nonetheless, we opted to keep the paper as accessible as possible for the broad readership of Climate of the past which is not necessarily expert in stratosphere dynamics.

*3. Section 4: First of all, the considered effects are not related to interactive chemistry but rather to the use of not appropriate ozone field. I guess, most of the differences discussed in this section disappear if the authors use the model w/o interactive chemistry, but with the ozone field prescribed from the Eocene run. I do not see any reason to compare Eocene run with the results of the model run driven by OzRoyer. Obviously, there will be substantial difference due to different situation during Eocene and present day. Comparison with Oz1855 is also not instructive because the ozone field is very close to the results of preindustrial run.*

We agree that the problem is the ozone field, that is exactly the point the paper is trying to make! In many paleo-climate modelling studies, the ozone field is simply a pre-industrial climatology. Our results suggest that it is not appropriate for Eocene-like conditions and has implications for climate. Our comparisons of the different simulations illustrate the sensitivity of the model-calculated middle atmospheric climate to the ozone field for Eocene conditions. In a way, using a pre-industrial ozone climatology is neglecting ozone chemical and dynamical feedbacks. A better way for having a more realistic ozone for Eocene climate is to have interactive chemistry in climate model, allowing to calculate an ozone field consistent with the paleo-climate. It's true that using a proper ozone climatology (i.e. calculated for Eocene conditions and ideally being a multimodel mean) could partly solve the problem. We do not argue that online coupling is required. We know this can be very difficult due to the computational cost of atmospheric chemistry and chemical tracer transport. In the abstract and conclusion of the paper that we recommend the use of suited ozone climatologies but not necessary an interactive chemistry climate model.

*4. Section 5: The problem here is related to the magnitude of radiative forcing. 1.8 W/m\*\*2 from stratospheric ozone increase looks extremely overestimated and has probably wrong sign. Forster et al., 2011 (doi: 10.1029/2010JD015361) showed using very accurate LBL radiation codes that 10% decrease of the stratospheric ozone gives only about 0.25 W/m\*\*2 (their Table 4). Portman and Solomon (2007, doi:10.1029/2006GL028252) concluded that the ozone radiative forcing caused by warming climate is within 0.1 W/m\*\*2. I think that very large 1.8 W/m\*\*2 ozone forcing (comparable to anthropogenic radiative forcing during 21st century) should be clearly explained. At least, its geographical, vertical and spectral signatures should be illustrated. In my opinion this forcing can be generated only by extraordinary high increase of the tropospheric or UTLS ozone (e.g., Beerling, 2011), which is not visible from presented results.*

In order to answer this comment and explain the 1.7W/m$^2$ radiative forcing (RF), we first explain briefly the set-up of the Forster et al., 2011 and Portman and Solomon (2007) studies and how our results are comparable (or not) with these studies. Then we discuss the climatic response in term of ozone RF and subsequent atmospheric changes in our set of simulations.

Portman and Solomon (2007) quantified the indirect effect of $CO_2$, $CH_4$, $N_2O$ increases, via stratospheric ozone changes. They use for that 2 projections of GHG for 2100 (scenarios SRES A2 and B1). In these scenarios, the GHG increases are far lower than the ones we consider. Based on these projections, they find radiative forcings due to ozone changes of -0.03 and +0.09 W/m\*\*2 depending on the scenario (their Table 2). It means that the sign of the RF due to stratospheric ozone change induced by simultaneous increase of several GHG is not obvious because the effect of $CO_2$ competes with the effect of $CH_4$ and $N_2O$, in particular in the altitude of ozone change (see their figure 2, 3 and 4). Thus, based on this study, it is not possible to argue that the sign of RF is obvious under warmer climate.

Forster et al. 2011 assessed the effect of an uniform 10% stratospheric ozone depletion for pressures less than 150hPa and find a positive RF of 0.25 W/m\*\*2 (resulting from a negative RF, -0.094 W/m\*\*2, in the longwave and a positive, 0.34 W/m\*\*2, in the shortwave). Bekki et al. 2013 used the same radiative model and also found that the longwave and short wave RF are both strongly affected by stratospheric ozone changes: They show that the ozone depletion in the tropical lower atmosphere leads to a positive forcing in the tropics due to a dominant positive RF in the short wave. However, based on multi-model climatologies from the future projections of CCMVal-2 exercises, Bekki et al. 2013 have also shown that this positive RF due to the depletion of ozone in the lower tropical stratosphere compete, in the tropics, with the negative RF due to ozone increase in the upper stratosphere when considering future climate and GHG conditions (see their Figure 2). Considering both lower and upper stratosphere, the RF is then negative in the tropics. In addition the ozone response to GHG and climate changes (and its subsequent RF) are of opposite signs in the tropics and extratropics (see their Figure 2) resulting finally in a positive multi-model mean RF of +0.131 W/m\*\*2 whereas RF for individual model projections lie in the -0.001 to +0.268 W/m\*\*2 interval. Therefore, it is not possible to assess the sign and amplitude of RF from stratospheric ozone changes without the use of a radiative model because of the extreme sensitivity of ozone RF to the altitude and latitude of ozone changes and the resulting competing effects in the SW and LW. This point is also highlighted by Bekki et al. 2013 (see their section [10]) who found a poor correlation between the global mean stratospheric ozone change and the RF.

Rev1 Figure 5 hereafter presents the distribution of ozone changes in order to facilitate comparisons with those discussed in Figure 2a of Bekki et al. 2013. In Bekki et al. 2013, the changes of ozone mass (between 2000 and 2100) reaches +40% for large part of the stratosphere, with maximum increase in the upper stratosphere and extra-tropical lowermost stratosphere. In the tropical lower stratosphere, the ozone decrease peaks at 20%. Under Eocene conditions, we calculate here ozone

increases in the upper stratosphere exceeding 40% and reaching up to 60-70%. The depletion of tropical ozone in the lower stratosphere is also much higher, exceeding 60%. In addition, the structure of the ozone changes differs from Bekki et al. ozone changes with an ozone decrease in the polar UTLS but a substantial increase in the tropical troposphere and subtropical barrier region. Our ozone changes are also, in magnitude, well beyond those presented in Figures 2 to 4 of Portman and Solomon (2007) which culminate in a 30% increase at 40km as a consequence of $CO_2$ increase for all the latitudinal band and an 8% decrease at about 24km for tropical latitudes. The differences in latitudinal mean profile in our simulation reach -50% for the 15°N and 15°S bands in the lower stratosphere and for the 75°N and 75°S bands in the upper troposphere. The ozone increase in the lower stratosphere at 45°N and 45°S reaches 40%. To conclude, the amplitude of the radiative forcing found in this study is greater than those reported in the publications mentioned by the reviewer since ozone changes are generally much higher and also differ in structure. As discussed in section 3.1, such large stratospheric ozone changes are not surprising for such a hot climate if we compare with studies on 4xCO2 climate.

[Figure]

**Rev1 Figure 5.** Difference in O3 mixing ratio (%) between the Eocene Interactive O3 simulation vs the Eocene simulation using a 11 year mean climatology centered on 1855 (EOCENE- EOCENE_Oz1855).

The underlying changes in the SW and LW radiative forcing as a function of latitude are presented hereafter and are now added in the new version of the paper. For the tropics, RF is found to positive in both longwave and shortwave. Beyond 50°, the positive SW RF is partly counterbalanced by negative longwave RF.

Net =  1.69 W.m$^{-2}$

| | | Downward | Upward | Net | Net SW+LW |
|---|---|---|---|---|---|
| Top of Atmosphere | Shortwave | 0.00 | -1.00 | 1.00 | 1.80 |
| | Longwave | | -0.80 | 0.80 | |
| 200hPa | Shortwave | -0.20 | -0.73 | 0.53 | 1.94 |
| | Longwave | -0.11 | -1.30 | 1.19 | |
| Surface | Shortwave | 0.30 | | | |

**Rev1 Table 1.** Differences in radiative fluxes between the interactive stratospheric chemistry (EOCENE) and the simulations using a 1855 climatology (EOCENE-Oz1855)

[Figure]

**Rev1 Figure 6.** Differences in radiative fluxes as a function of latitude between the interactive stratospheric chemistry (EOCENE) and the simulations using a 1855 climatology (EOCENE-Oz1855)

Other methodological differences exist in the way we compute RF compared with Portman and Solomon (2007) and Forster et al. 2011. Here, we use the methodology explained p669 of the 5[th] WGI assessment report of the IPCC consisting on computing the difference of the net top radiative fluxes at TOA in two GCM simulations forced by sea surface temperatures. It means that we compute the ozone radiative forcing taking into account the fast tropospheric feedbacks (e.g. changes in temperature, clouds and humidity profiles). Therefore, our ozone RF includes some of its indirect effects whereas Portman and Solomon (2007), Forster et al. 2011 and Bekki et al. 2013 only discussed direct ozone RF. In addition, in Portman and Solomon (2007) and in Forster et al. 2011, 2-D ozone distributions (latitudinal means) are considered whereas we use 3-D distributions.

=> The Rev1 Table 3 and Rev1 Figure 6 are now presented in the paper

*The estimation of the surface temperature response using some other model sensitivity to homogeneous radiative forcing is oversimplified. If the obtained 1.8 W/m\*\*2 is true (which I doubt) it will show the importance of the problem by itself.*

As our aim is to show the importance of stratospheric ozone changes under Eocene conditions and their potential relevance for climate. We just want to know whether its climate effect is negligible or might be significant or even important compared to the effects of other parameters discussed in the literature. Our aim, by estimating rather crudely the temperature change, is not to be quantitative but to have a first estimation since stratospheric ozone changes and associated impact on surface climate have not been explored so far for Eocene conditions. Such estimation, even crude, allows to conclude on the potential importance of considering stratospheric ozone change in comparison to the change in boundary conditions.

---

## Author Comment (AC2) · 12 Feb 2019

We thank the **reviewer #2** for his/her comments, which will lead to a clearer manuscript notably regarding the seasonality of the changes in the stratospheric circulation.

Hereafter, we explain how we are able to improve the paper regarding the issues mentioned by the reviewer #2 in blue italic (actions taken on the manuscript are preceded by an arrow).

*The main motivation appears to be the climate sensitivity to stratospheric ozone, but this cannot be studied with the model setup used: because SSTs and other lower boundary conditions are prescribed, surface climate cannot respond to atmospheric changes. In fact, the prescribed SSTs, which come from a low-resolution coupled climate model (FOAM, which as far as I understand does not include interactive chemistry), are likely inconsistent with the ozone feedback that would result from the inter- active chemistry simulations with LMDz.*

Our main motivation is not the climate sensitivity to stratospheric ozone. The first focus is the stratospheric ozone changes themselves in the Eocene and then the associated first-order climate effects. If the study had been about climate sensitivity to stratospheric ozone, we would have ran 4xCO2 simulations with other boundary conditions being set at present-day conditions (to be comparable to most other studies on the topic) and if focused on climate sensitivity we would have tried to use a fully-coupled ocean-atmosphere model (as recommended by the reviewer hereafter). Since it is the first study on stratospheric ozone changes for Eocene, it makes sense to use an atmospheric climate model forced by SST to make a first estimation of ozone distribution changes, to explore the drivers (easier to do with a fixed-SST configuration than in a fully coupled configuration) and assess the potential climate forcing. Such configuration has been applied many time for such paleoclimate investigations [e.g. Botsyun et al. 2019, Ladant et al., 2014; Ladant et al. 2016; Licht et al., 2014; Pohl et al. 2016; Porada et al. 2016]. It is more reasonable to launch into fully coupled long simulations only if ozone changes and first-order effects are found to be potentially significant in the fixed-SST configuration.

Our ultimate objective here is to estimate the first-order climate signal that can be missed in a typical warm paleoclimate simulation when the response of stratospheric ozone to Eocene conditions and associated dynamical feedbacks are ignored. This first-order impact is the ozone-driven changes in atmospheric dynamics, temperature and radiative balance. As noted by rev#2, our ocean is not interactive, so we missed the effect on sea-surface temperatures, and the associated potential feedbacks. We consider that it is not necessary to include the ocean feedback, which requires a far more complex model setting and longer computation times, for an estimation of first-order effects.

**Ref:**

Botsyun S., P. Sepulchre, Y. Donnadieu, C. Risi, A. Licht, J. K. Caves Rugenstein. "Revised paleoaltimetry data show low Tibetan plateau elevation during the Eocene.", Science (2019), *in press*.

Ladant, J. B., Y. Donnadieu, V. Lefebvre, and C. Dumas (2014), The respective role of atmospheric carbon dioxide and orbital parameters on ice sheet evolution at the Eocene-Oligocene transition, Paleoceanography, 29, 810–823, doi:10.1002/2013PA002593.

Ladant, J.-B. & Donnadieu, Y. Palaeogeographic regulation of glacial events during the Cretaceous supergreenhouse. Nat. Commun. 7:12771, doi: 10.1038/ncomms12771 (2016).

Licht, A., et al. (2014), Asian monsoons in a late Eocene greenhouse world, Nature, 513(7519), 501–506.

Pohl, A., Y. Donnadieu, G. Le Hir,J.-B. Ladant, C. Dumas, J. Alvarez-Solas,and T. R. A. Vandenbroucke (2016), Glacial onset predated Late Ordovician climate cooling,Paleoceanography,31, 800–821,doi:10.1002/2016PA002928.

Porada, P. et al. High potential for weathering and climate effects of non-vascular vegetation in the Late Ordovician. Nat. Commun. 7:12113 doi: 10.1038/ncomms12113 (2016).

*Given this issue, another motivation is to study the stratospheric response (ozone, circulation) to the prescribed Eocene conditions. This would be fine, but in the current presentation in the manuscript this appears to be poorly conceived:*

*a) Only annual mean cross sections are shown, even though the essential dynamics take place in each hemisphere's winter and spring seasons. I think that seasonal- mean plots are necessary to substantiate the results and interpretations related to the stratospheric response.*

As pertinently pointed out by the reviewer, in the stratosphere, essential dynamics take place in hemisphere's winter season, when the polar vortex dominates the large scale zonal circulation at mid-and-high latitudes. Initially, the annual mean plots were meant to provide a general overview of the major stratospheric changes on ozone and circulation in a standard way. In addition, timeseries in key sectors (e.g. zonal-mean zonal wind in the climatological center of the NH polar night jet (60°N,10hPa), meridional eddy heat flux at 100 hPa and 45-75°N) provided valuable information on the seasonal evolution of the stratospheric dynamics, essentially to better characterize and understand the total column ozone (TCO) gradient which appears peculiar in our study in comparison with other studies where present-day climate $4xCO_2$ experiments are analyzed. Nonetheless, we agree that providing additional analysis of the seasonal evolution of the stratosphere dynamics using monthly or seasonal mean cross-section can help elucidate mechanisms, notably and help to (i) understand the apparent contradiction noticed by the reviewer of a faster Brewer-Dobson circulation despite a stronger polar vortex, and (ii) understand better the TCO gradient in the Northern Hemisphere. Therefore, the analysis is now extended by adding Northern Hemisphere's (where changes are the strongest) monthly mean zonal cross sections of the zonal-mean wind, transformed eulerian mean diagnostics, etc, for preindustrial conditions and the anomalies associated with Eocene conditions. Please find more details in our response to comment e).

=> In the revised version of the manuscript, an entirely new analysis of the seasonal stratosphere dynamics - based on seasonal mean cross section in Northern Hemisphere winter as suggested by the reviewer – has been performed and presented section 3.2 which has been renamed "Seasonal evolution of the Northern Hemisphere stratospheric polar vortex in Eocene conditions". Figures 5 and 6 have been removed and replaced by the three Figures shown in our response to comment e). Figure 5 has been moved in the supplementary material as we believe that it illustrates very clearly the seasonal changes of the polar vortex.

*b) Changes to tropospheric CH4 and N2O are applied but no results are presented that show what the effects of these changes are (N2O should lead to modified chemistry, as noted by authors; CH4 could lead to changes in stratospheric H2O). If these changes are deemed to not be so important then why not simply perform simulations with 4xCO2, which would also have the advantage of providing better comparability to previous results with similar forcing? If above changes of tropospheric species are deemed to be important then this calls for corresponding analyses and results to be presented.*

[Figure]

**Rev2 Figure 1.** Changes in zonal-mean $O_3$ (in ppmv), $H_2O$ (in%), temperature (in K) and zonal wind (in m/s) associated with the changes of $N_2O$ and $CH_4$ under Eocene conditions. Dotted region indicates that the anomalies are not statistically significant at the 5% level.

As shown above (Rev2 Figure 1), changes in $CH_4$ and $N_2O$ lead to changes in $H_2O$ and $O_3$ that are small but consistent with the expected response (less than 5% increase of stratospheric water vapour due to increasing methane and a similar max ~5% decrease of ozone due to increase $N_2O$). The associated changes in dynamics appear also to be small (see zonal mean annual temperature and January zonal mean zonal wind on Figure 1). In the case of the zonal wind, the changes are insignificant for the January month. Other months show some patches of statistical significance but longer runs would be required to assess the robustness of these dynamical changes. In summary, these changes are small compared to the anomalies that are initially discussed in the paper; i.e. EOCN-PREIND and anomalies related to ozone mis-specification.

=> In the revised version of the manuscript, we now put more emphasis on the above mentioned anomalies (which are more important for the climate community), rather than adding an extra discussion on the role of $CH_4$ and $N_2O$ that appear to be of secondary importance in light of the results presented above and whose effects are not very important, particularly with regard to the uncertainties of the climate 55 Ma years ago. Nonetheless, the effect of $N_2O$ and $CH_4$ will be mentioned and included in the associated plots as supplementary material. We modified the manuscript as follows:

"Note that in comparison with a standard $4xCO_2$ simulation, including a 17% increase of $N_2O$ in our Eocene simulations leads to a slight decrease of ozone which reaches a maximum of 3% in the equatorial upper stratosphere (5 hPa) (see supplementary material). Although the $N_2O$ increase influence on ozone is statistically significant, its impact appears to be small compared to the 40% upper stratosphere ozone increase due increasing $CO_2$."

*c) How well does this model (LMDz) simulate the stratospheric circulation compared to other state-of-the-art chemistry-climate models? The CCMVal-2 activity included a version of this model, which*

*indicates it is performing well in terms of several diagnos- tics, but also has some issues (see SPARC CCMVal report referenced): e.g., huge warm bias in upper stratosphere, where the radiative scheme seems to behave questionably, bias in surface energy balance, large cold bias in SH leading to strongest ozone hole out of all compared models etc. The only place in the literature where I could find a plot of the model's overturning streamfunction (i.e., its BDC) is Dietmueller et al. 2018 (Fig. 1 therein): it looks completely off, questioning the model's ability to simulate stratospheric transport (despite the fact that its AoA distribution looks okay) . . . in all fairness, Dietmueller et al. note that this may be a diagnostic, rather than an actual model problem. In any case, the authors should include information about the basic model performance in regards to stratospheric dynamics, transport, and climate, and convince the reader that this is a suitable model for the purposes of the study.*

The model has been involved in a range of studies, model inter-comparisons and evaluation. The overall conclusion is that LMDz is not the best or worst chemistry-climate model, it all depends on the chosen diagnostics and the selected regions. On a more general level, is there anything to gain in having all of us running the same model with the same set up, unless we want to end up with the same results? Below, we show several results of recent literature where the middle atmosphere dynamics of LMDz (or LMDz-Reprobus) have been evaluated against various reanalysis and other models in the frame of CCMI and other projects (e.g. CMIP). This should give an overview of how LMDz performs. It appears from the first example (Rev2 Figure 2 taken from de la Camara et al. [2016a]) that the stream function and zonal-mean zonal wind (here shown in DJF) – despite some differences - are overall highly consistent in LMDz and ERA-I reanalysis. In this case, LMDz is far from being completely off.

[Figure]

**Rev2 Figure 2.** Zonally averaged zonal wind (in m $s^{-1}$, shaded), and stream function of the residual mean meridional circulation (contours) in LMDz. Magenta contours represent positive values (i.e., clockwise circulation), and cyan contours represent negative values (i.e., counter-clockwise circulation). From de la Camara et al. (2016a)

Both figures below (Figure 3, taken from de la Camara et al. [2016b]) give an overview of the seasonal evolution of the Southern Hemisphere polar vortex in LMDz in comparison with ERA-I. Again, despite some differences, the model appears to perform reasonably well, also near the stratopause and above. Interestingly, the final warming date (onset of the Southern Hemisphere polar vortex break-up) appears particularly well represented (comparing the blue and red vertical profile on Figure 4) in LMDz. LMDz shows only few days delay in the climatological final warming date; this cold bias appears to be particularly small with regard to the large interannual variability. Note that this reduced cold bias in the Southern Hemisphere is, to a large extent, due to the advanced non-orographic wave parameterization, which, if inaccurately specified (see de la Camara et al. [2016b] for more details) leads to an actual polar vortex break-up delay of ~15 days.

[Figure]

**Rev2 Figure 3.** Time–height evolution of (a) zonal-mean zonal wind (m s$^{-1}$) averaged over 70°–50°S and (b) temperature (K) averaged over 85°–60°S during the southern winter and spring for ERAI. (c),(d) As in (a) and (b), respectively, but for LMDZ. From de la Camara et al. (2016b)

[Figure]

**Rev2 Figure 4.** Final warming dates as a function of pressure level in ERAI (1992–2011; red), LMDZ (blue), and LMDZ-CS (green). The climatological means are given by the solid lines, and the shaded areas represent plus or minus one standard deviation. From de la Camara et al. (2016b).

Finally, the Northern Hemisphere stratospheric winter time variability also appears to be of good quality as revealed by the results of Ayarzagüena et al. [2018]. In their study, they showed that, first, the Sudden Stratospheric Warmings (SSWs) mean frequency in CCMI LMDz-REPROBUS is close to the one derived from reanalysis. Among CCMI models, LMDz further appears to be one of the closest to the real world. As shown below (Rev2 Figure 5), the mean duration (partly radiatively driven) and deceleration of the PNJ and their standard error in LMDz are particularly consistent with reanalysis. LMDz appears to perform well compared to other models that participated to CCMI.

[Figure]

[Figure]

**Rev2 Figure 5**. (a) Duration of SSWs (in days) and (b) deceleration of the PNJ associated with SSWs (in m s$^{-1}$) in each model for both periods of study. Bars denote ±1.5 standard error, and green stars indicate future values that are statistically significantly different from the past ones at the 95 % confidence level. From Ayarzagüena et al. (2018).

These various results demonstrate that the LMDz model simulates the middle atmosphere dynamics and circulation decently. Of course, the comparison with reanalysis is not perfect, but never completely off. Furthermore, these results demonstrate that LMDz performs reasonably well in comparison with other models of the same kind. In the revised version of the manuscript, we now recall this by citing relevant examples.

The focus by the reviewer on the Dietmueller et al. 2018 reference is very unfortunate and rather selective. How is it possible to reconcile the "completely off" model's overturning streamfunction (i.e., its BDC) shown in Dietmueller et al. 2018 with all the other results (and not only the AoA distribution)? Dietmueller et al. concluded: " The reason for these additional circulation cells is unknown. However, as the model shows a reasonable AoA, there might be a diagnostic problem in the residual circulation data". They are right. The reviewer also seems to have serious doubts about this plot. Unfortunately, Dietmueller et al. had contacted the wrong person for our CCMI runs, instead of the CCMI PIs (S. Bekki and M. Marchand), contrary to the CCMI guidelines, and no LMDz people are co-authors of this study. We are now in touch with Dietmüller et al. to sort it out, perhaps including a correction to the publication. At this stage and "in all fairness", the LMDz performances should be assessed in the light of all the other studies, including CCMI and CMIP inter-comparisons

=> In the revised version, additional information on model performances are now included in section 2.1.

**Ref:**

de la Cámara, A., Lott, F., and Abalos, M.:Climatology of the middle atmosphere in LMDz: Impact of source-related parameterizations of gravity wave drag, J. Adv. Model. Earth Sys., 8, 1507– 1525, https://doi.org/10.1002/2016MS000753, 2016a.

de la Cámara, A., F. Lott, V. Jewtoukoff, R. Plougonven, and A. Hertzog (2016b), On the gravity wave forcing during the southern stratospheric final warming in LMDz, J. Atmos. Sci., 73, 3213–3226, doi:10.1175/JAS-D-15-0377.1

Ayarzagüena, B., Polvani, L. M., Langematz, U., Akiyoshi, H., Bekki, S., Butchart, N., Dameris, M., Deushi, M., Hardiman, S. C., Jöckel, P., Klekociuk, A., Marchand, M., Michou, M., Morgenstern, O., O'Connor, F. M., Oman, L. D., Plummer, D. A., Revell, L., Rozanov, E., Saint-Martin, D., Scinocca, J., Stenke, A., Stone, K., Yamashita, Y., Yoshida, K., and Zeng, G.: No robust evidence of future changes in major stratospheric sudden warmings: a multi-model assessment from CCMI, Atmos. Chem. Phys., 18, 11277-11287, https://doi.org/10.5194/acp-18-11277-2018, 2018.

*d) Other studies on ozone changes due to 4xCO2 (see references listed above) have highlighted the crucial role of changes in stratospheric H2O, which come about due changes in tropical tropopause temperature, but also due to ozone-temperature feedbacks near the tropical tropopause. This type of*

*sensitivity could be important in order to understand the climate response to 4xCO2 and should be included in the results and discussion.*

We agree that stratospheric $H_2O$ changes could play a significant role in the climate feedbacks and help to understand the climate response to $4xCO_2$. Note that it is not only stratospheric $H_2O$ but also changes in high altitude clouds (cirrus). These water cycle feedbacks may explain some of the large model and scenario dependency of climate impacts associated with ozone changes (Nowack et al., JGR, 2018). However, we think the issue of the water cycle changes should be explored in a less constrained configuration (i.e. coupled atmosphere-ocean framework). In addition, tackling this issue is a paper in itself and is beyond the scope of the present paper.

*e) At face value, the presented results indicating both an accelerating BDC and stronger polar vortex seem to contradict each other, since a stronger BDC should be associated with stronger wave drag, which would be consistent with a weaker vortex. This is not discussed in the paper but seems important to understand the stratospheric changes. My guess is that this can be explained by the seasonality in the changes (cf. Figs. 5, 6): the wave forcing seems indeed weaker in early winter when the vortex is much stronger (and I would expect a weaker BDC during that part of the season, but this should be checked and potentially included in the presented results). During late winter and spring the wave forcing is much enhanced consistent with an accelerated BDC - again this should be checked based on residual circulation diagnostics.*

We thank the reviewer for pointing out this apparent contradiction and his/her very relevant insights. As shown on Figure 6 below (which is now inserted in the revised version of the manuscript and which is consistent with the zonal-mean zonal wind seasonal evolution shown a 10 hPa/60°N in the former version of the manuscript – Figure 5), a much stronger stratospheric polar vortex develops in early winter (Nov-December) under the Eocene climate compared to pre-industrial conditions. The strength of the polar night jet is doubled over the full depth of the stratosphere. By late winter (starting in January), the anomalies progressively reverse from the upper part of the stratosphere. In March, the stronger polar vortex anomalies in the middle atmosphere is no longer significant. As noticed by the reviewer, such a strong polar vortex anomaly seems at first glance in contradiction with a faster Brewer-Dobson circulation. Analysis of the wave activity and its interaction with the mean flow (i.e. engine of the brewer Dobson circulation or extratropical pump) allows removing this apparent contradiction.

[Figure]

**Rev2 Figure 6**. Seasonal evolution (Oct to Mar) of the zonal mean zonal wind differences between the Eocene and preindustrial conditions. Shaded contours indicate that anomalies are significant at the 5% levels according to a t-test. Black isocontour shows the preindustrial run climatology.

[Figure]

**Rev2 Figure 7**. Seasonal evolution (Oct to Mar) of the Eliassen-Palm Flux (vectors) and its divergence (contours, in m/s/d) under preindustrial conditions.

The preindustrial climatology of the planetary wave propagation (EP-flux) and its interaction with the mean flow (EP-Flux divergence) shows that, permanently in winter, the wave activity penetrates in the stratosphere and the breaking of planetary waves lead to westward momentum drag which maximize near the location of the southern flank of the polar night jet (Rev2 Figure 7). This contributes to erode and weaken the polar vortex, to a warming of the polar stratosphere and lead to a net poleward residual mass transport (which drives the Brewer-Dobson circulation to a large extent). The wave activity and its interaction with the mean flow peaks in January but is already large in November (which can eventually lead to SSW [e.g. de la Camara et al., 2016]). Note that these model results are very consistent with reanalysis (see also response to comment c)) and therefore indicate that the representation of the stratosphere dynamics and circulation in LMDz-Reprobus is of an overall good quality.

[Figure]

**Rev2 Figure 8**. Seasonal evolution (Oct to Mar) of the differences between the Eocene and preindustrial conditions of the Eliassen-Palm Flux and its divergence. Shaded contours indicate that anomalies are significant at the 5% levels according to a t-test. Preindustrial climatology is shown with dashed contours.

Under Eocene conditions (Rev2 Figure 8), it appears that the planetary wave penetrating the stratosphere in early winter (Nov-Dec) is significantly reduced and deflected equatorward as revealed by the downward and equatorward pointing of the EP-Flux vector in the lower mid-latitude stratosphere (see also Figure 6b showing a lower eddy heat flux at 100 hPa in the former version of the manuscript). This is associated with an anomalous positive E-P flux divergence throughout the depth of the stratospheric polar night jet (near 60°N), which indicates a reduced westward momentum forcing by waves and hence allows a stronger development of the polar vortex in early winter. In contrast, by January, the wave activity becomes significantly larger (see also Figure 6b), the westward forcing appears strongly amplified in the upper stratosphere and this momentum forcing anomaly progressively propagates downward. This is consistent with the reversal of the zonal mean zonal wind anomaly in the upper stratosphere, but also with the overall extremely rapid deceleration of the polar vortex strength seen on Figure 5. Relatively, the Brewer-Dobson circulation will hence be less reduced in early winter than accelerated in late winter (where the differences in the wave forcing are much stronger), which results in a net acceleration of the Brewer-Dobson under Eocene conditions compared to preindustrial conditions as revealed by the younger age of air. Note that we also examined the residual circulation velocities v* and w* which confirms the seasonal changes in the Brewer-Dobson circulation strength (not shown).

=> This analysis is now included in the revised version of the paper in section 3.2. Accordingly, section 3.2 has been renamed "Seasonal evolution of the Northern Hemisphere stratospheric polar vortex in Eocene conditions". Figures 5 and 6 have been removed and replaced by the three Figures shown in our response to comment e). Figure 5 has been moved in the supplementary material as we believe that it illustrates very clearly the seasonal changes of the polar vortex.

**Ref:**

de la Cámara,A., Lott, F., and Abalos, M.:Climatology of the middle atmosphere in LMDz: Impact of source-related parameterizations of gravity wave drag, J. Adv. Model. Earth Sys., 8, 1507– 1525, https://doi.org/10.1002/2016MS000753, 2016.

*f) It is claimed that the stratospheric cooling due to higher CO2 levels explains the changes in polar vortex strength, but why would the CO2 cooling affect the meridional temperature gradient rather than lead to a meridionally uniform cooling, which would not affect the polar vortex? With the presented results the cause of the strengthened vortex remains confusing.*

We thank the reviewer for noticing this flawed explanation. This comment has prompted us into analyzing more thoroughly the mechanism. Indeed, the $CO_2$ cooling in the stratosphere should be uniform and hence should therefore not affect the meridional temperature gradient. As shown on Figure 1b of the former manuscript, however, the increase of ozone in the upper stratosphere could inflect further the shortwave heating rates gradient on the winter hemisphere. This is indeed what we found with an overall more negative meridional SW heating gradient throughout the depth of the stratosphere which maximizes at the stratopause. This effect is however balanced by the fact that equatorial temperatures are decreasing at a faster rate than the polar temperatures in the Eocene simulation. So the net difference is slightly positive in the middle stratosphere and negative at the stratopause level. Radiative effects on the polar vortex appear hence to be modest compared to the changes in the wave activity described in the previous comment. The manuscript has been revised accordingly.

=> confusing statements regarding $CO_2$ radiative effect is now removed in the revised version of the manuscript (section 3.2 and conclusion) and a discussion related to changes in the wave activity and the stratospheric wave drag (based on the diagnostics shown in the response to comment e)) is now added.

*Minor Comments:*

*Fig. 1: why not present AoA similar to panels a, b (difference as color shading with PI control as black contours)?*

[Figure]

**Rev2 Figure 9.** Age of air (contour lines) calculated after 20 years of simulations by taking as a reference entry point the equatorial lowermost stratosphere, slightly above the tropopause (i.e. pressure level corresponding to 74 hPa). Shaded contour shows the difference between the Eocene and preindustrial experiments.

Interestingly, it appears that the Brewer-Dobson acceleration is more intense in the Northern Hemisphere, consistently with the wave-mean flow interaction diagnostics.

=> This age of air figure now replaces Figure 1c of the former manuscript.

*Also: what are the units for the presented PI O3?*

=> This is now clarified.

*page 4, section 2.1: it would help to include some information about how the model compares to other chemistry-climate models (see major comment)*

=> Some elements have been added in the revised version of manuscript based on the discussion of the major comment c).

*page 4, line 31: please also provide the model top*

=> We now mention the model top (~70 km) *"15 levels above 20 km and around 24 above 10km and a lid height at ~70 km"*.

*page 5, line 3: "snapshots" - do you mean "time slices"?*

=> yes, this is now modified in the revised version

*page 5, line 17: please explain "LPJ"*

=> LPJ is the Lund-Potsdam-Jena vegetation model. This is now clarified.

*page 5, line 24: please provide justification / motivation for why you choose a CO2 value at the low end of what's recommended*

As explained earlier, we used an existing protocol for Eocene to show for the first time the potential importance of the stratospheric ozone feedback for past climate simulations. We will continue to investigate this issue with more complete set of simulations in the future (which will arise from the DeepMIP protocol).

*page 6, line 1-2: please provide more detailed explanation for why radiative effect due to enhanced CH4 and N2O levels would be accounted for by enhanced CO2?*

Even if inferred from proxies, the temperature changes for Eocene are better known than the greenhouse gases content (for which only $CO_2$ level can be inferred but with large uncertainties). For this reason, paleoclimate modellers have investigated the Eocene climate running simulations with various $CO_2$ covering a large range of concentrations with the aim of reproducing the amplitude of surface temperature changes. They do not change the level of other GHGs because no data are available on them. However, as only $CO_2$ is adjusted to match the temperature difference it means that the $CO_2$ perturbation implicitly represents the sum of all the GHG perturbations. This methodology is the one recommended by Lunt et al. 2017 for the DeepMIP project. For that reason we only perturb $CO_2$ in FOAM and LMDz for Eocene simulations.

Thus, In the LMDz-REPROBUS chemistry-climate modelling, fixed $CO_2$, $CH_4$ and $N_2O$ are used as inputs to the radiative scheme. As a result, only ozone changes influence the climate during a preindustrial or Eocene simulation. That way, the effect of ozone changes on middle atmospheric climate can be isolated and quantified. Obviously, ozone chemistry is also affected by changes in $N_2O$ and $CH_4$, two key stratospheric source gases. To account for this effect, there are $CH_4$ and $N_2O$ chemically active tracers (i.e. modified by the transport and chemistry schemes) and whose surface concentrations are taken from the modelling study of Beerling et al. 2011. Their global distributions change with time during a simulation, but they are not used as inputs to the radiative scheme and hence their changes do not affect the climate, only ozone changes do.

=> we have now clarified this point in section 2. 2 and in the Table 1

*page 6, line 10: "80s" - you mean the 1980's? Is this meant to represent an "ozone-hole climate"? Why not simply use the O3 field from, e.g., a CCMVal-2 chemistry-climate simulation with your model?*

Indeed, we mean 1980's, it is now clarified in the text. We describe here what is available in our atmospheric model.

*page 6, line 25: please discuss the temperature changes a bit more, e.g.: is the Antarctic amplification (largest temperature response over Antarctica) a well-known response for these types of simulations? Why is there no corresponding Arctic amplification as happens for current climate change and happens for pure 4xCO2 runs?*

In present climate conditions, the polar amplification is well more pronounced in the Arctic than in the Antarctic. This has been the subject of many studies and several key processes have been identified; i.e. the surface albedo feedback [e.g. *Serreze and Francis*, 2006] (increase in surface absorption of solar radiation due to snow and ice retreat), temperature feedbacks [e.g. *Pithan and Mauritsen*, 2014] or changes in poleward heat transport [e.g. *Graversen et al.*,2008], etc. Large uncertainties however remain in quantifying the contribution of these various processes. Under Eocene climate, EOMIP simulations from various Earth-System model analyzed by *Lunt et al*. [2012] revealed a stronger Antarctic than Arctic amplification, consistently with proxy records. The greatest warming in the Antarctic region is due to the lower topography (see Figure below) via the lapse rate effect and the change in albedo. The role of topography on Antarctic amplification was further demonstrated by *Sazlmann* [2017].

=> A short mention of this has been added in section 3.1 of the revised version of the manuscript, when Figure 1 is described.

**Ref:**

Serreze, M. C., and J. A. Francis (2006), The arctic amplification debate, Clim. Change, 76, 241–264.

Pithan F and Mauritsen T (2014), Arctic amplification dominated by temperature feedbacks in contemporary climate models, Nat. Geosci. 7 181–4

Graversen, R. G., Mauritsen, T., Tjernstrom, M., Kallen, E., and Svensson, G.: Vertical structure of recent Arctic warming, Nature, 451, 53–56, 2008.

Lunt, D. J., Dunkley Jones, T., Heinemann, M., Huber, M., LeGrande, A., Winguth, A., Loptson, C., Marotzke, J., Roberts, C. D., Tindall, J., Valdes, P., and Winguth, C.: A model–data comparison for a multi-model ensemble of early Eocene atmosphere–ocean simulations: EoMIP, Clim. Past, 8, 1717-1736, https://doi.org/10.5194/cp-8-1717-2012, 2012.

Salzmann, M.: The polar amplification asymmetry: role of Antarctic surface height, Earth Syst. Dynam., 8, 323-336, https://doi.org/10.5194/esd-8-323-2017, 2017

*page 7, line 13: "total ozone column" - here and elsewhere: usually this is referred to as "total column ozone (TCO)" and I'd recommend nomenclature consistent with other literature*

=> done

*page 8, line 3, Fig. 4: you already showed an indication that the winter season matters most, so why not show DJF and JJA changes instead of the annual mean (see major comments)?*

=> section 3.2 of the manuscript has been revised accordingly (and deeply). See also response to major comments.

*page 9, line 1: "... drives the strength of the zonal wind" - 1) thermal wind balance doesn't tell you about cause and effect, so "drives" is misleading, 2) it's a relation between the meridional temperature gradient and the vertical zonal wind gradient (not the wind itself), so you wouldn't necessarily expect the temperature gradient at 10 hPa to correspond to the wind at 10 hPa . . .*

We agree, this is misleading.

=> The part on this has been removed in the revised version and section 3.2 is now entirely revised.

*page 9, line 2: the heat flux is a proxy for the vertical Eliassen-Palm (~wave activity) flux, which more accurately also involves the vertical temperature gradient and the background vorticity; given that you compare two very different climates, I wonder whether the heat flux is a sufficiently accurate measure of wave activity flux, since both background temperature and vorticity structures might contribute?*

The vertical component of the EP flux is expressed as

$$F_z = \rho_0 a \cos \Phi \, f \, \frac{\overline{v'\theta'}}{\partial \bar{\theta}/\partial z}$$

where $\rho_0$, $a$, $\Phi$, $f$ and $\Theta$ are the air density, earth radius, latitude, Coriolis parameter and potential temperature, respectively. Under Eocene conditions, the planetary vorticity could have been slightly different as the Earth was rotating faster (the length of the day would have been less than an hour less corresponding to less than 4% difference), but this is not accounted for in our simulations. The vertical gradient of the potential temperature in the stratosphere increases as a result of higher $CO_2$ but the difference is small and almost null at 100 hPa, where we calculated it. The difference is hence expected to be modest. As shown by the comparison on Rev2 Figure 10 below, the heat flux appears to be sufficiently accurate (note that the eddy heat flux is multiplied by $\cos\Phi$). Finally, note that the meridional eddy heat flux term is, per definition, a quantification of the departure from the zonal mean which characterizes the wave itself, so this appears to be the major term in the measure of the wave activity.

[Figure]

**Rev2 Figure 10.** (left) vertical component of the Eliassen-palm flux at 100 hPa and averaged in the 45-75°N latitude band for (black) Preindustrial and (red) Eocene experiments. (right) same as left but for the meridional eddy heat flux.

---

## Author Comment (AC3) · 12 Feb 2019

We are thankful to R. Orbe for this careful review of our manuscript. His/her comments are very positive. We hope the next manuscript will be even clearer.

---

## Author Response (AR1)

Dear Editor,

Following your recommandations, we made major corrections and added new figures to our manuscript in order to answer the reviewer's questions. Please find hereafter a marked-up manuscript version. Regarding more specifically the major issues you highlighted (in blue) we did the following changes:

1. The used model setup, in particular the forcing boundary condition for the FOAM simulation and for the simulations in this study, should be more clearly presented. The prescribed Eocene SST from FOAM does not allow feedbacks with a coupled-ocean system. It also need to be compared to existing model intercomparison projects like Lunt et al., 2012 (Climate of the Past). In addition, it should be clearly addressed how successful FOAM is to simulate the deep past climate.

Regarding the model setup, we added a figure (Figure 1) to clarify the protocol to produce boundary conditions. We also added in the text a comparison between our SST with the ones presented in Lunt et al. 2012 and added recent references based on the same protocol which highlight its robustness for paleo simulations (when a fully coupled climate model can not be used). See p.6 of the version showing the changes.

2. Seasonal mean plots are necessary to better asses interpretations related to the stratospheric response.

In the revised version of the manuscript, an entirely new analysis of the seasonal stratosphere dynamics - based on seasonal mean cross section in Northern Hemisphere winter as suggested by the reviewer 2– has been performed and is presented section 3.2 which has been renamed "Seasonal evolution of the Northern Hemisphere stratospheric polar vortex in Eocene conditions". Figures 5 and 6 from the first manuscript have been removed and replaced by the three Figures 6, 7 and 8. The ex-Figure 5 has been moved in the supplementary material (now Figure S2) as we believe that it illustrates very clearly the seasonal changes of the polar vortex.

3. The role of change in tropospheric CH4 and N2O and their effects on stratospheric chemistry should have been discussed in more detail. It may require additional sensitivity experiments as suggested by one reviewer.

As shown in the answer to reviewer 2 (Rev2 Figure 1), the changes in $CH_4$ and $N_2O$ lead to changes in $H_2O$ and $O_3$ that are small but consistent with the expected response (less than 5% increase of stratospheric water vapour due to increasing methane and a similar max ~5% decrease of ozone due to increase $N_2O$). The associated changes in dynamics appear also to be small. In summary, these changes are small compared to the anomalies that are initially discussed in the paper; i.e. EOCN-PREIND and anomalies related to ozone mis-specification.

In the revised version of the manuscript, we now put more emphasis on the anomalies induced by ozone mis-specification (which are more important for the climate community), rather than adding an extra discussion on the role of $CH_4$ and $N_2O$ that appear to be of secondary importance in light of the results presented above and whose effects are not very important, particularly with regard to the uncertainties of the climate 55 Ma years ago. Nonetheless, the effect of $N_2O$ and $CH_4$ are mentioned and included in the associated plots as supplementary material. We modified the manuscript as follows:

"Note that our simulation, the stratospheric chemistry is also modified by the increase of $N_2O$ and $CH_4$. However, their effect only reaches a maximum of 3% in the equatorial upper stratosphere (~5 hPa) (see supplementary Figure S1). Although this chemical effect on ozone is statistically significant, its impact appears to be small compared to the upper stratosphere 40% increase in ozone due to increasing $CO_2$."

4.      The revised manuscript need to consider an improved discussion about how the LMDz simulate the stratospheric circulation compared to other state-of-the-art chemistry-climate models.

In the revised version, additional information on model performances are now included in section 2.1.

5.      The strong acceleration of the boreal polar night jet, which is more than two times stronger during Eocene, does not agree with previous publications and requires a more detailed discussion.

In the revised version of the manuscript, the section 3.2 dealing with the seasonal stratosphere dynamics analysis has been renamed "Seasonal evolution of the Northern Hemisphere stratospheric polar vortex in Eocene conditions" and totally rewritten to fully explain the changed in polar jet.

Sincerely

Sophie SZOPA

[revised manuscript text omitted]

Sophie Szopa 11/2/y 11:51

Supplementary Figures for the paper **"Role of the stratospheric chemistry-climate interactions in the hot climate conditions of the Eocene"** by Sophie Szopa, R. Thiéblemont, S. Bekki, S. Botsyun, P. Sepulchre

[Figure]

**Figure S1:** Annual mean differences (EOCENE experiment (EOCNclimchim) with $N_2O$ and $CH_4$ changed minus EOCENE experiment (EOCNclim) with $N_2O$ and $CH_4$ at preindustrial levels in the chemistry model) of zonally averaged ozone (in vmr). Color filled contours indicate that anomalies are statistically different at the 1% confidence level according to a t-test. Black contours show the EOCNclimchim climatology expressed in vmr.

[Figure]

**Figure S2:** Time series of the monthly mean zonal-mean zonal wind at 60° N and 10 hPa (~31 km) for the Eocene and preindustrial simulations. Red and black curves show the climatological averages for the Eocene and Preindustrial simulations, respectively. All simulated years are individually plotted in thin yellow (EOCENE) and grey (PREIND). Note that each year is repeated twice to visualize better the seasonal cycle.